# Site-specific photolabile roadblocks for the study of transcription elongation in biologically complex systems

Jean-François Nadon[1,6], Vitaly Epshtein[2,3], Etienne Cameron[4,6], Mikhail R. Samatov [5], Andrey S. Vasenko [5], Evgeny Nudler[2,3] & Daniel A. Lafontaine [1✉]

Transcriptional pausing is crucial for the timely expression of genetic information. Biochemical methods quantify the half-life of paused RNA polymerase (RNAP) by monitoring restarting complexes across time. However, this approach may produce apparent half-lives that are longer than true pause escape rates in biological contexts where multiple consecutive pause sites are present. We show here that the 6-nitropiperonyloxymethyl (NPOM) photolabile group provides an approach to monitor transcriptional pausing in biological systems containing multiple pause sites. We validate our approach using the well-studied *his* pause and show that an upstream RNA sequence modulates the pause half-life. NPOM was also used to study a transcriptional region within the *Escherichia coli thiC* riboswitch containing multiple consecutive pause sites. We find that an RNA hairpin structure located upstream to the region affects the half-life of the 5′ most proximal pause site—but not of the 3′ pause site—in contrast to results obtained using conventional approaches not preventing asynchronous transcription. Our results show that NPOM is a powerful tool to study transcription elongation dynamics within biologically complex systems.

[1] Department of Biology, Faculty of Science, RNA Group, Université de Sherbrooke, Sherbrooke, QC J1K 2R1, Canada. [2] Department of Biochemistry and Molecular Pharmacology, New York University School of Medicine, New York, NY 10016, USA. [3] Howard Hughes Medical Institute, New York University School of Medicine, New York, NY 10016, USA. [4] Department of Chemical Engineering, Polytechnique Montreal, Montreal, QC H3T 1J4, Canada. [5] HSE University, 101000 Moscow, Russia. [6]Present address: Pancosma Canada Inc, Drummondville, QC J2C 7V5, Canada. ✉email: daniel.lafontaine@usherbrooke.ca

Transcriptional pausing is essential in prokaryotes and in eukaryotes[1–5]. In *Escherichia coli*, it is observed that pause sites occur at approximately every 100 nucleotides throughout the genome[6]. Pausing in bacteria plays a central role in biological processes such as the coordination of transcription and translation[7], RNA folding[8,9], cotranscriptional sensing[10–13], and Rho transcription termination[12,14]. In eukaryotes, transcriptional pausing is found to maintain RNA polymerase II in the vicinity of promoters to ensure a rapid response following sensing of extracellular growth factors[15,16]. Pausing of RNA polymerase II is also important to keep an active chromatin architecture, to be critical for coordinating rapid tissue development in *Drosophila*[17,18], and to be involved in the differentiation of mouse and zebrafish embryonic stem cells[19–21]. Clearly, transcriptional pausing is extensively involved in genetic regulation mechanisms occurring in all living organisms.

Kinetic models describe transcriptional pausing as an elongating complex (EC) entering an unproductive state that may be caused by numerous factors such as structural rearrangements, reverse translocation, or RNA polymerase-nascent RNA contacts[22]. In vitro characterization of transcriptional pausing relies on kinetic assays where RNAP complexes escaping from a single pause site are modeled to an exponential decay function to determine the half-life ($t_{1/2}$) of the pause (Fig. 1a, top panel; Supplementary Methods)[23]. In such a case, the half-life of a pause site is directly affected by variations in EC escaping rates (Fig. 1b)[24]. However, the measurement of the pause half-life may yield an apparent dwell time that is longer than the true pause escape rate in the presence of multiple consecutive pause sites[25–27]. In this context, incoming EC from upstream pause sites may increase the EC occupancy of downstream sites (Fig. 1a, bottom panel). It is anticipated that variations (10- to 30-folds) in the rate of incoming EC could significantly affect the apparent half-life of downstream pause sites when fitting the data with a single-exponential decay (Fig. 1c; Supplementary Methods). Importantly, multiple consecutive pause sites are observed in several biological systems[9,10,12,28], suggesting that asynchronous transcription in such contexts may complicate the determination of the actual rate of pause escape and therefore the biological relevance of transcriptional pause sites.

Nucleic acid scaffolds may be used to reconstitute transcriptional complexes to study the *his* pause RNA hairpin[29]. Although such scaffolds are ideal to specifically introduce structural probes within the RNA sequence, they are limited to the study of a single pause site and may prove challenging to reconstitute EC harboring long RNA fragments (>30 nucleotides). Other methods employing biotin-streptavidin complexes[12,30–32] and EcoRI (Q111) roadblocks are employed to synchronize ECs at specific pause sites by sterically blocking elongating RNAP[33]. However, since the biotin-streptavidin complex is highly stable[34], it is not practical to release stalled RNAP without disrupting elongation complexes. Although the EcoRI (Q111) roadblock is readily removable using high salt concentrations, it is expected that high salt concentrations—and the added EcoRI restriction site sequence—are most probably affecting nascent RNA structures. In addition, it is possible to use the catalytically dead *Streptococcus pyogenes* CRISPR-Cas9 enzyme (dCas9) to block *E. coli* RNAP elongation[35]. However, since the roadblock only partially stalls *E. coli* RNAP elongation, its use to study pause site escape is compromised. Lastly, a promising new chemical transcriptional roadblocking system is also available[36] but due to the permanent nature of the compound, its use is restricted to the study of stalled complexes.

The compound 6-nitropiperonyloxymethyl (NPOM) is a photocaging group[37] that inhibits DNA base pairing when incorporated in a thymine base (Fig. 1d). However, the use of UVA light selectively removes NPOM from DNA, thus allowing thymine to perform base pairing interactions (Fig. 1d)[38–41]. Importantly, the use of UVA light to remove NPOM allows to probe biological processes without altering native properties of biomolecules[42–44]. NPOM can be used for the modulation of transcription initiation by selectively allowing access to the DNA by the nuclear factor κB[45] or triplex-forming oligonucleotides[46]. NPOM may also be used to block the DNA extension of oligonucleotides by the T4, Taq, and Phusion DNA polymerases[47]. Based on these results, we reasoned that NPOM could be used as a transcriptional roadblock to stall EC at specific pause sites, effectively allowing the synchronization of transcriptional complexes. Then, upon exciting NPOM with UVA light, we hypothesized that NPOM removal

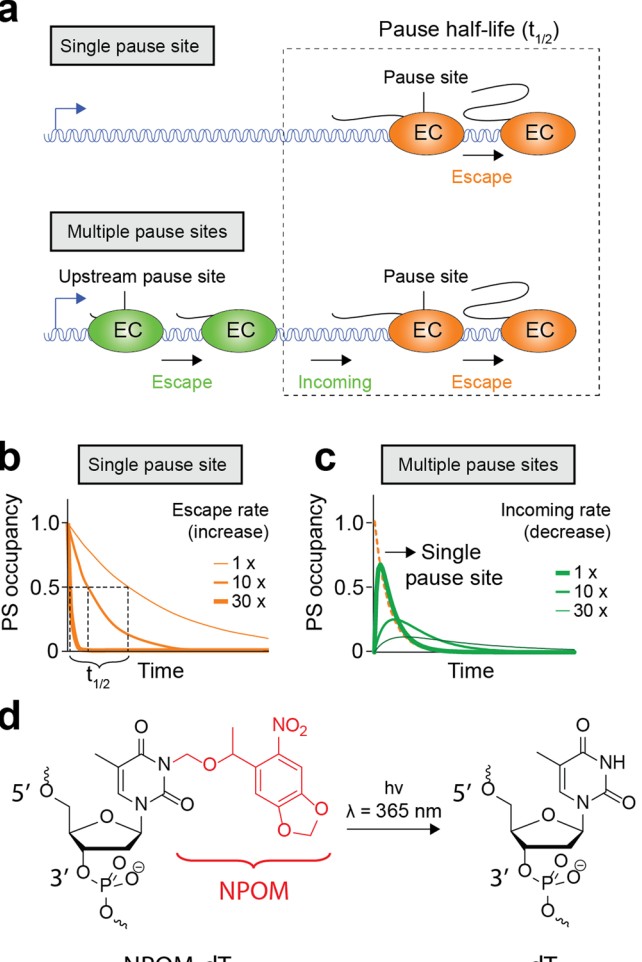

**Fig. 1 Transcriptional pausing in the context of single and multiple pause sites. a** Schematics of single (top panel) and multiple (bottom panel) pause sites. The pause half-life is characterized by the rate of EC escape from the pause site (indicated by dotted rectangle). In the case of multiple pause sites, EC escaping from upstream sites may enter in downstream pause sites. **b** Simulations of single pause site occupancy by EC as a function of time. The rate of pause escape is inversely proportional to the pause half-life, as observed by the faster decrease of pause site occupancy when increasing the rate of EC escape. **c** Simulation of multiple pause sites occupancy by EC as a function of time. The variation in upstream pause escape modulates the observed half-life of a downstream pause. It can be observed that decreasing rates of upstream pause escape increases the observed half-life of the downstream pause (green curves). The occupancy of a single pause site is shown as reference (orange curve). **d** Chemical structure of an NPOM-caged dT and an unmodified dT following cleavage by ultraviolet (λ = 365 nm) irradiation.

would permit stalled EC to be released from the pause site and therefore to directly measure the pause half-life by monitoring escaping ECs. Due to the stepwise synchronization and release of elongating ECs, NPOM is expected to provide an approach to quantify the half-life of transcriptional pausing.

In this work, we characterized NPOM as a removable transcriptional roadblock. We found that NPOM efficiently stalls elongating *E. coli* RNAP one residue upstream of the NPOM incorporation site. We observed that NPOM-stalled ECs remain stably formed even when exposed to extensive washing steps. Our results also showed that UVA light removes NPOM-roadblocking groups and that unrestricted ECs may resume transcription elongation upon the addition of nucleotide triphosphates (NTP). No evidence of RNAP backtracking was detected when subjecting NPOM-stalled ECs to the GreB transcription factor. As a control, we used NPOM to monitor the well-studied *his* pause site. We found that the pause half-life obtained using NPOM was very similar to that obtained using synchronized transcription reactions, indicating that NPOM can be used to reconstitute ECs that are transcriptionally paused. We also used NPOM to elucidate the role of a hairpin structure in the *E. coli thiC* riboswitch that was previously reported to modulate the half-life of multiple consecutive downstream pauses. Using NPOM-modified templates, we found that the half-life of the 3′ pause site is not directly affected by the hairpin structure. Instead, our results show that the half-life of the 3′ pause site is artificially increased by the presence of asynchronous ECs escaping the upstream pause site. Together, our data demonstrate that NPOM can be used to unambiguously study the half-life of pause sites within biologically complex regulatory systems relying on transcriptional pausing to coordinate gene expression.

## Results

**NPOM efficiently blocks elongating *E. coli* RNAP.** We first prepared DNA templates containing the NPOM group to be used for in vitro transcription reactions. To do so, we engineered a DNA construct containing a *lacUV5* promoter and an NPOM-caged dT nucleotide incorporated at position 90 relatively to the transcription start site (TSS) (Fig. 2a). Due to the presence of NPOM, it is expected that no base pair is formed between the NPOM-containing nucleotide and the corresponding A90 residue (Fig. 2a). The transcript sequence was derived from the *tbpA* riboswitch[48] and an extension of five nucleotides was added to the riboswitch 5′ region to enhance transcription efficiency (Supplementary Fig. 1a, b). As NPOM was previously reported to block PCR amplification[47], a primer extension approach using an NPOM-caged DNA primer was employed to prepare the DNA template (Supplementary Fig. 2a). After the elongation of the NPOM-containing primer, electrophoretic mobility shift assays (EMSA) showed that the double-stranded NPOM DNA template was readily obtained (Supplementary Fig. 2b).

To assess the ability of NPOM to block transcription elongation, we performed in vitro transcription experiments using the *E. coli* RNAP. When transcription reactions were performed using a DNA template not containing an NPOM group, a unique transcript species corresponding to the full-length sequence was observed (Fig. 2b). However, when employing an NPOM-containing DNA template, a shorter RNA species was detected after a 10 min transcription reaction (Fig. 2b), suggesting that NPOM successfully blocks RNAP elongation. Furthermore, only ~1 to ~3% of transcription readthrough could be observed when incubating transcription reactions for longer periods up to 1 h (Fig. 2b), consistent with NPOM completely blocking RNAP progression. RNA sequencing transcription reactions performed using 3′-O-methyl nucleotides revealed that

NPOM caused ECs to stall at position G89 (Fig. 2c), which corresponds to one nucleotide prior to the NPOM-caged dT. This finding is consistent with data obtained for the T4, T7, Taq, and Phusion DNA polymerases indicating that polymerization is arrested one residue prior to the NPOM-incorporated site[47]. These results can be explained by the fact that NPOM is a relatively small chemical group that might be accommodated by the RNAP catalytic site, thus preventing transcription elongation through the inhibition of Watson–Crick base pairing with incoming ATP. Alternatively, it is also possible that NPOM-dT fails to translocate into the active site, which would result in transcripts having the same 3′ end in halted complexes.

To determine whether transcription elongation complexes remain intact when stalled by NPOM—and that they do not dissociate from DNA—his-tagged RNAP was immobilized on Ni-NTA agarose beads and was used to produce NPOM-stalled ECs. In these assays, it is expected that nascent transcripts remain associated to RNAP after a washing step if NPOM-stalled ECs remain intact. Upon washing complexes, a large fraction of transcripts was still detected in the RNAP fraction after extensive washing, suggesting that ECs remain stably formed when stalled by NPOM (Fig. 2d, NPOM). Very similar results were obtained when using the high affinity biotin-streptavidin complex as a transcription roadblock (Fig. 2d, Strept)[12,31], suggesting that NPOM is as efficient as the biotin-streptavidin complex to yield stable ECs. Control experiments showed that no run-off transcript remained bound to the RNAP after washing in the absence of a transcriptional roadblock (Fig. 2d, run-off). Together, these results show that ECs remain stably bound to the DNA template when encountering the NPOM transcriptional roadblock.

It was previously shown that the biotin-streptavidin complex may lead to RNAP backtracking when used as a transcriptional roadblock[31]. In such a case, NPOM would be hard to use as a transcriptional roadblock for the study of transcriptional pausing as it would leave an uncertainty relatively to the RNAP position. Thus, to test whether NPOM leads to RNAP backtracking, NPOM-stalled ECs were incubated with the GreB transcription elongation factor that cleaves extruded RNA 3′ ends from backtracked ECs (Fig. 2e)[49]. When the biotin-streptavidin roadblock was used to stall ECs, cleaved roadblock (RB) transcripts were detected (Fig. 2e, Streptavidin), as expected from backtracking occurring in this context. No such cleaved transcripts were observed when NPOM was used to induce EC stalling (Fig. 2e, NPOM), suggesting that ECs do not backtrack upon encountering the NPOM roadblock. In both cases, GreB cleavage activity was found to occur on transcription initiation complexes (Fig. 2e, see backtracked initiation complexes), consistent with backtracking taking place within initiation complexes. Thus, our results show that NPOM roadblocking does not induce ECs to undergo backtracking. Importantly, although these results show that NPOM and streptavidin produce different backtracking effects, further experiments are required to determine whether backtracking is either inhibited with NPOM or promoted with streptavidin.

**Removal of NPOM allows stalled RNAP to resume elongation.** We next investigated the efficiency of UVA light to remove NPOM from the DNA template (Fig. 3a). We first performed a time-dependent exposure of NPOM-containing DNA templates to UVA light and subsequently used the resulting templates for in vitro transcription assays. As expected, while the presence of the NPOM roadblock is associated with strong RNAP stalling, UVA illumination of NPOM-containing DNA templates leads to the transcription of the full-length RNA species (Fig. 3b). In

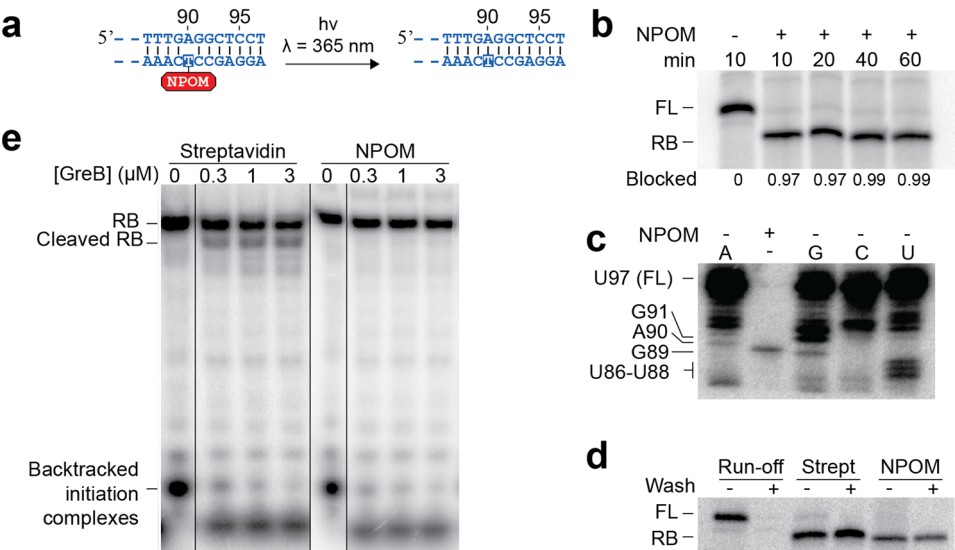

**Fig. 2 Characterization of NPOM as a transcriptional roadblock. a** Schematic representation of the DNA template used for the characterization of NPOM as a transcriptional roadblock. Only the 3′ extremity of the template is shown. NPOM-dT has been incorporated on the template strand at the 90th position. Upon UVA exposure, NPOM is cleaved, thus yielding an unmodified dT. **b** In vitro transcription experiments performed in the absence (−) or presence (+) or NPOM. Transcription was allowed to proceed for indicated amounts of time. The full-length (FL) and roadblocked (RB) species are shown. **c** Mapping of NPOM-roadblocked transcripts using 3′-OMe RNA sequencing reactions. The 3′-OMe nucleotide is shown above each lane and the nucleotide sequence is shown on the left. **d** Transcription assays performed in presence of streptavidin and NPOM roadblocks that were treated without (−) and with (+) washing steps. ECs were bound to Ni-NTA agarose beads for washing steps. Run-off transcription was used as a negative control. **e** NPOM-roadblocked and biotin/streptavidin-roadblocked ECs were subjected to GreB before and after UV irradiation. As a positive control, internal biotin/streptavidin roadblocked templates were used[31]. Roadblock (RB) and cleaved roadblock transcripts (cleaved RB) are shown on the left. The experiments were performed at least two times and variations were less than 5%.

particular, these results revealed that NPOM was removed from ~85% of DNA templates only after 8 min of UVA illumination (Fig. 3b). Control experiments performed in the absence of NPOM showed that transcription initiation complexes (EC14) exposed to UVA light for an identical time (8 min) resumed elongation very efficiently (~85%) (Supplementary Fig. 3). Together, these experiments suggest that UVA light efficiently removes NPOM from DNA templates while minimally perturbing elongation complexes.

In addition to efficiently blocking elongating RNAP, it is crucial for NPOM to be readily removed by UVA even when engaged within roadblocked ECs. To investigate this, we prepared NPOM-stalled ECs and subjected them to UVA light to determine the efficiency of transcription elongation restart. In these assays, the removal of NPOM should allow ECs to resume elongation due to the presence of unincorporated NTPs in the reaction. UVA illumination of NPOM-stalled ECs revealed that a large proportion (~73%) of ECs resumed elongation (Fig. 3c), consistent with NPOM being efficiently removed from DNA even in the presence of stalled ECs.

The sequence specificity of transcription restart of NPOM-stalled ECs was next investigated using stepwise transcription reactions. In these experiments, NPOM-stalled ECs were passed through gel filtration columns to wash unincorporated NTP and were subsequently exposed to UVA light to remove the NPOM roadblock. Importantly, given that ECs are expected to stall at position G89, the addition of ATP allowed the incorporation of a single nucleotide, thus resulting in transcripts being extended at position A90 (Fig. 3d, ATP). However, no elongation was observed when using either UTP, CTP, or GTP (Fig. 3d). A longer transcript species was observed when adding a mixture of ATP and GTP, corresponding to position G92 (Fig. 3d). In all cases where transcription restart was monitored, elongation efficiencies reached ~50% (Fig. 3d) suggesting that a fraction of

ECs are not in an active conformation to restart elongation. Although more work is required to characterize this process, G50 columns used for washing steps during EC preparation may lead to a fraction of complexes not resuming transcription elongation. Thus, our data show that stalled ECs that are obtained following NPOM removal efficiently resume transcription elongation and are performed as expected from the template sequence.

NPOM was also tested to determine its ability to roadblock the bacteriophage T7 RNAP during elongation. To do this, we produced DNA templates with and without NPOM and in which the T7 RNAP promoter was incorporated. Transcription reactions performed using the T7 RNAP showed that NPOM effectively blocks RNAP elongation (Fig. 3e, + NPOM). However, NPOM-removal using UVA illumination did not yield full-length transcripts (Fig. 3e, + UVA), which is in contrast to experiments performed using the E. coli RNAP (Fig. 3c). Our results suggest that although NPOM successfully blocks transcription elongation by the T7 RNAP, the UVA-dependent removal of NPOM does not allow ECs to resume transcription elongation. Additional work will be needed to determine whether stable T7 RNAP complexes are obtained through NPOM roadblocking or if such complexes are unstable.

It was previously reported that NPOM could be used to modulate polymerase chain reaction reactions through the selective binding of NPOM-containing DNA primers[41,50]. To determine if NPOM could also be used to selectively control DNA polymerization through roadblocking, we performed primer extension assays using the Taq and Phusion DNA polymerases. Upon allowing both DNA polymerases to extend a radioactively labeled DNA primer, we observed that the extension was blocked by NPOM in both cases (Fig. 3f), in agreement with previous findings[47]. Importantly, when removing NPOM using UVA, we found that both DNA polymerases resumed polymerization (Fig. 3f). These results suggest that the elongation process of Taq

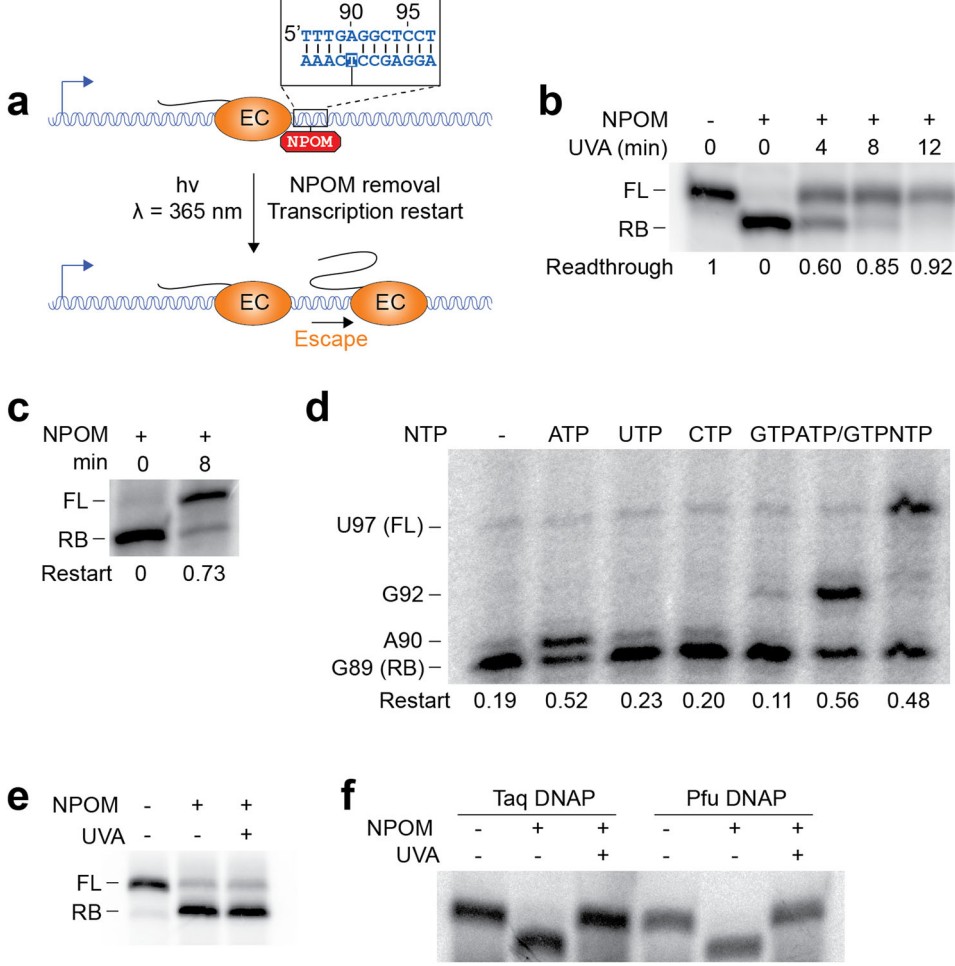

**Fig. 3 NPOM removal allows transcription elongation to resume. a** Transcriptional roadblocking by NPOM, followed by transcription restart upon UV irradiation. **b** NPOM-caged DNA templates were subjected to UV light ($\lambda = 365$ nm) for increasing amounts of time, before transcription. **c** NPOM-roadblocked ECs were subjected to UV light with NTPs still present in the reaction mixture. Production of the full-length transcript indicates that transcription resumes after NPOM cleavage. **d** NPOM-roadblocked templates were purified on a G50 column and water (−) or ATP, UTP, CTP, GTP, ATP, and GTP or NTPs (50 μM) were added before ultraviolet irradiation. Transcription resumes after the addition of ATP, but not UTP, CTP, or GTP. When ATP was present (A, AG, and NTP), ultraviolet exposition allowed transcription to resume up to positions A90, G92, and U97, respectively. **e** NPOM efficiently blocks transcription by the T7 RNAP, but does not allow restart upon UV irradiation. Transcription was performed using a template containing the T7 promoter, and a GCG upstream of the 5′ GUU. **f** NPOM efficiently blocks replication by the Taq and Pfu DNAPs. NPOM cleavage by UV irradiation allows elongation to resume. NPOM removal was performed in the presence of nucleotides for panels (**c**–**f**). The experiments were performed at least two times and variations were less than 10%.

and Phusion DNA polymerases may be selectively controlled through NPOM roadblocking activity. Thus, our data show that NPOM can be employed to synchronize and monitor specific steps of coreplicational regulatory events such as G-quadruplex formation[51].

**Use of NPOM to study the *his* pause half-life.** To demonstrate the use of NPOM to monitor transcriptional pausing and the associated half-life, NPOM was employed to study the well-characterized *his* pause[52]. The *his* pause is a non-backtracked pause that is stabilized through interactions occurring between RNAP and the nascent RNA hairpin, which is constituted by five base pairs and a loop of eight nucleotides (Fig. 4a)[53]. The *his* pause was also shown to be increased through the hairpin by the elongation factor NusA[10,52–57].

We designed templates for in vitro transcription reactions allowing to produce the first 132 nucleotides of the *his* transcript in which the pause site is located at position U102 (Fig. 4a, full-length). We also constructed a minimal template not containing

most of the sequence located upstream of the *his* RNA hairpin (Fig. 4a, minimal length). We first monitored the kinetics of transcription elongation in the absence of NPOM in the context of the full-length construct. In these experiments, we first generated EC20 complexes by omitting UTP in the transcription reactions. Following a washing step, we then added NTPs and monitored transcription elongation at different time points (Supplementary Fig. 4). We observed that the *his* pause exhibited an apparent half-life of $37 \pm 5$ s (Fig. 4b and Supplementary Table 1), which is similar to previous data[29,52]. To allow elongating complexes to stall at the *his* pause site (U102), we next prepared a G103A mutant DNA template to incorporate NPOM. No significant effect of the G103A mutation was observed on the apparent half-life ($44 \pm 2$ s) when compared to the wild-type sequence (Supplementary Table 1). Control transcription reactions showed that the NPOM-containing template efficiently stalled EC (Supplementary Fig. 5), consistent with the presence of NPOM. However, following UVA light exposure, high yield of full length transcripts was observed as

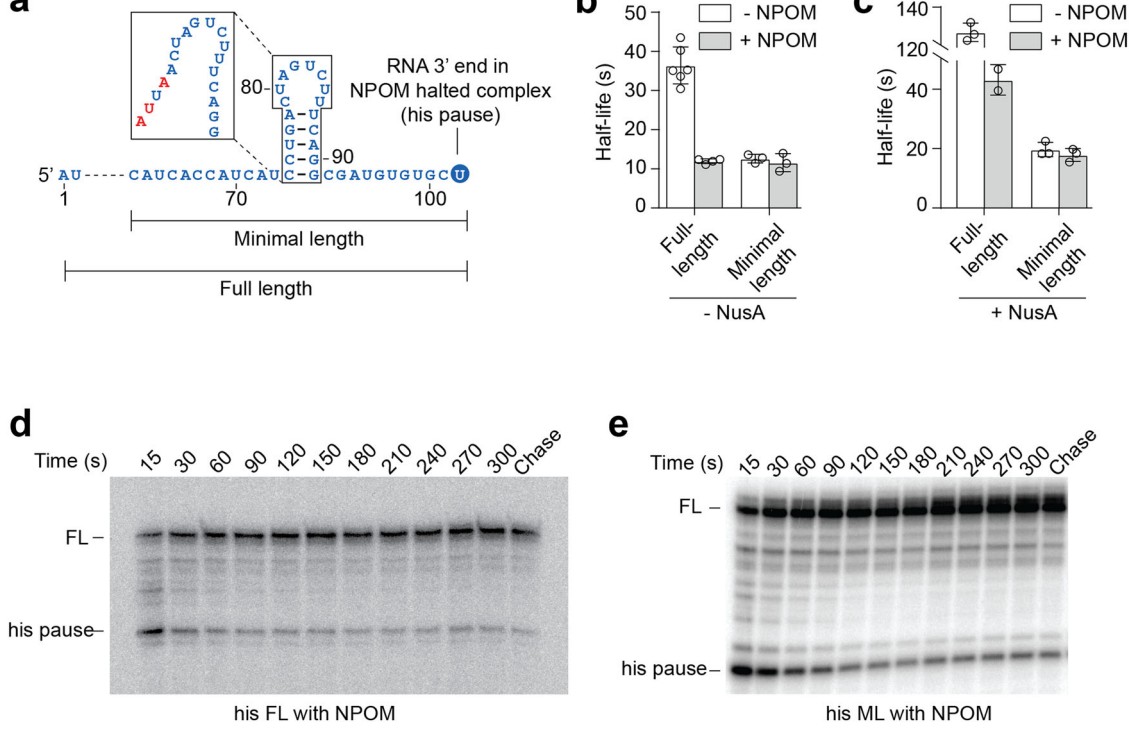

**Fig. 4 EC synchronization at the pause site using NPOM allows observation of the *his* pause. a** The different templates that were used allowed transcription of the entire *his* transcript (Full length, FL) or only a small section containing the pause hairpin (Minimal length, ML). **b** Observed half-life values of the *his* pause using the various constructs in the absence of NusA. The average values of independent experiments with standard deviations (SD) are shown. **c** Observed half-life values of the *his* pause using the various constructs in the presence of NusA. **d** Transcription kinetics experiment using the full-length (FL) *his* template with ECs synchronized at the *his* pause using a NPOM roadblock. **e** Transcription kinetics experiment using the minimal length (ML) *his* template with ECs synchronized at the *his* pause using a NPOM roadblock. The pause site and full-length species are shown. NPOM removal was performed in the absence of NTPs for panels (**d**, **e**). The experiments were performed at least two times and variations were less than 10%.

expected from the removal of NPOM. Using this technique to synchronize EC at the *his* pause site, we monitored EC escaping from the *his* pause upon adding NTP. We observed that released EC exhibited a shorter pause half-life ($12 \pm 1$ s) (Fig. 4b, d and Supplementary Table 1), which is ~3.5-fold lower than the half-life measured using the non-NPOM template (Fig. 4b). We reasoned that the shorter pause half-life was caused by the NPOM synchronization of ECs at the *his* pause site, thereby resulting in the half-life strictly representing pause escape (Fig. 4b). This is in contrast to transcription reactions performed without NPOM where incoming asynchronous ECs may continuously populate the *his* pause site during half-life measurement, most likely resulting in a larger value ($44 \pm 2$ s) not representing the intrinsic pause half-life.

Using our approach to synchronize EC at the *his* pause site through NPOM, we next assessed the importance of the *his* hairpin for the pause half-life by disrupting the hairpin structure through mutations (Fig. 4a, see insert)[52]. Transcription kinetics experiments showed that the first time point already contained a near maximal amount of full-length transcripts (Supplementary Fig. 6), suggesting that the pause is severely affected when the hairpin is mutated. These results indicate that ECs escaping the pause site after NPOM removal are still regulated by the hairpin-stimulated *his* pause, which is in good agreement with previous results[53].

According to our data, reducing the upstream sequence of the *his* pause site should decrease the proportion of upstream asynchronous incoming EC, which is expected to reduce the apparent half-life. In agreement with this, transcription reactions of templates harboring a smaller transcript (Fig. 4a, minimal

length, ML) yielded a shorter apparent pause half-life ($13 \pm 1$ s) (Fig. 4b, e). Interestingly, a nearly identical value ($12 \pm 2$ s) was obtained using an NPOM-containing minimal length construct (Fig. 4b, e and Supplementary Table 1). These results are in agreement with the idea that asynchronous ECs do not artificially increase the *his* pause half-life in the smaller construct. Since the half-lives calculated for the full-length and minimal constructs are determined using different initially halted complexes (EC20 vs EC102), it is possible that the respective rate of escape of these complexes may contribute to some extent to the observed differences in pause half-life.

**Use of NPOM to investigate the role of NusA**. We next employed our approach to study the influence of NusA on ECs escaping the *his* pause site[52,57,58]. When transcribing the full-length *his* transcript in the presence of NusA, we measured an apparent half-life of $127 \pm 5$ s (Fig. 4c and Supplementary Table 1). This result indicates that NusA is increasing by ~3.5-fold the *his* pause half-life, as previously reported[29,52]. Using NPOM-containing templates, we allowed ECs to restart from the *his* pause site in the presence of NusA and obtained a half-life of $44 \pm 5$ s (Fig. 4c and Supplementary Table 1). When comparing to the value obtained without NusA ($12 \pm 1$ s) (Supplementary Table 1), it suggests that NusA binding to ECs specifically slows pause escape by ~3.6-fold. Our data are in good agreement with a previous study using reconstituted nucleic acid scaffolds where NusA was found to affect EC pause escape[29]. Thus, our results show that NPOM may be used to directly monitor the effects of NusA on pause escape without interference from asynchronous elongations.

It has been previously reported that NusA may contact RNA upstream of the *nut* hairpin[59], likely through its extended RNA binding surface[60]. To investigate whether the upstream RNA sequence of the *his* pause is important for NusA-dependent pause escape, we next used the NPOM approach to monitor transcription reactions using the minimal length template. In the presence of NusA, we determined half-lives of $20 \pm 2$ and $18 \pm 2$ s (Fig. 4c) using templates without and with NPOM, respectively, which represents a half-life increase of ~1.5-fold compared to values obtained in the absence of NusA (Fig. 4b). Thus, in the context of NPOM templates where only the *his* pause escape is monitored because NusA had a larger effect on full-length (~3.6-fold) vs minimal length (~1.5-fold), it suggests that NusA modulates the pause half-life at least partly by interacting with upstream RNA sequence. This is consistent with the NusA positively charged flank, which constitutes an RNA binding surface allowing continuous scan for regulatory structures[61].

**Study of transcriptional pausing within a multiple consecutive pause sites region**. Several complex biological systems contain consecutive pause sites[9,10,12,28]. One such system is found in the *E. coli thiC* 5′ untranslated region, which contains a riboswitch domain regulating the expression of proteins involved in TPP biosynthesis[62]. In the riboswitch, three consecutive pauses have been observed at positions A138, C158, and C187[12,63], the latter acting as a regulatory checkpoint for gene expression[12]. In such a complex regulatory system, it is possible that EC escaping the A138 pause may enter the downstream C158 pause, which would artificially increase the observed half-life of the pause C158 (Fig. 5a, top panel). We recently showed through mutational analysis that the half-life of the A138 pause is decreased when preventing the formation of an upstream hairpin structure (Supplementary Fig. 7a)[28]. Unexpectedly, it was observed that the A138 hairpin mutant also decreased the apparent half-life of the C158 pause[28]. These results could indicate that the C158 pause half-life is decreased due to faster incoming of upstream ECs that are escaping from the weaker A138 pause site (Fig. 5a, middle panel).

To investigate how the A138 hairpin modulates the half-life of the C158 pause, we employed the NPOM approach to specifically monitor the rate of C158 pause escape. We first designed templates to transcribe a section of the *thiC* riboswitch encompassing the A138 hairpin and both A138 and C158 pause sites. Sequencing transcription reactions revealed that both pause sites were preserved in the context of this construct (Supplementary Fig. 7b). We determined that the apparent pause half-lives of A138 and C158 were of $14 \pm 3$ and $26 \pm 7$ s, respectively (Supplementary Table 1), consistent with previously obtained data using the full-length riboswitch[28]. To synchronize transcription complexes at the C158 pause, we next verified that the use of a G159A variant required for NPOM attachment did not affect transcriptional pausing at positions A138 and C158 ($16 \pm 2$ and $27 \pm 2$ s, respectively) (Supplementary Table 1). Lastly, we also engineered a disrupted A138 hairpin construct and assessed both pause half-lives, which yielded values of $8 \pm 1$ and $16 \pm 4$ s for A138 and C158, respectively. Thus, our results indicate that the disruption of the A138 hairpin decreased the half-lives of both A138 and C158 pauses, as previously observed in the context of the full-length riboswitch[28].

We next prepared a similar template in which NPOM was incorporated at the position G159A to synchronize ECs at the C158 pause site (Fig. 5a, bottom panel). Control experiments showed that while NPOM-containing DNA templates efficiently cause roadblocking, the use of UVA light resulted in the transcription of the full-length species (Supplementary Fig. 5c).

Using the NPOM template to synchronize ECs at the C158 pause site, we next monitored pause escape by adding NTP and determined a pause half-life corresponding to $16 \pm 3$ s (Fig. 5b and Supplementary Table 1). Strikingly, when performing a similar experiment in the context of a A138 hairpin mutant, we determined that the C158 pause half-life corresponded to a value of $18 \pm 3$ s (Fig. 5b, c and Supplementary Table 1). This value is very similar to what observed using the wild-type A138 hairpin construct ($16 \pm 3$ s), suggesting that the stability of the A138 hairpin does not modulate the rate of C158 pause escape. Together, our data indicate that the use of NPOM may allow us to remove the indirect effect of the A138 hairpin on the C158 pause half-life[28].

The results obtained here clearly suggest that the apparent dwell time of RNAP at the C158 pause site depends on ECs escaping both A138 and C158 pauses. In such a situation, it is possible to estimate the half-life of the C158 pause by performing numerical analysis using a two-step kinetic model. Using our data obtained in the absence of NPOM, we used the equation of Bateman to calculate the half-life of the C158 pause using a two-step kinetic model taking into account incoming ECs (see "Methods"). When performing these calculations using the wild-type and the A138 hairpin mutant, we found that the C158 pause half-life was not significantly changed ($16 \pm 4$ s and $15 \pm 5$ s, respectively). Thus, these results support our NPOM data suggesting that the apparent half-life of the C158 pause is modulated by incoming ECs from the A138 pause, which are modulated by the integrity of the A138 hairpin structure.

## Discussion
Numerous factors such as incoming elongation complexes and the rate of pause escape may influence the calculated half-life obtained from traditional transcription kinetics experiments. Half-life measurements rely on the prior synchronization of EC upstream to the pause site by omitting a nucleotide from the transcription reaction, thus resulting in halted elongation complexes. This is important to minimize the upstream continued arrival of complexes at the pause site, which could introduce errors in calculating the rate of pause escape and pause half-life[23]. Nucleic acid scaffolds and artificially stalled complexes have been routinely used to synchronize EC upstream of pause sites and both approaches are expected to yield relevant data when using relatively short transcribed templates. However, in transcribed regions spanning several hundreds of nucleotides and containing multiple consecutive pause sites—as observed in biologically complex systems[6,12,64,65]—it may be practically impossible to accurately determine the pause half-life when using conventional approaches due to ECs asynchronization. In principle, although it could be attractive to remove upstream RNA stretches and/or extraneous pause sites to obtain better EC synchronization, this would result in non-native transcripts being produced, therefore potentially overlooking the influence of upstream RNA in the transcription assays. Furthermore, the removal of upstream pause sites could induce misfolded RNA as previously observed in ribozymes and riboswitches[1]. Importantly, even when not altering the transcribed region of interest, prior experimental evidence suggests that transcriptional asynchronization occurs in presence of at least two consecutive pauses, which leads to a biased interpretation of a hairpin-dependent pause site in the *thiC* riboswitch[28]. It is expected that biological systems relying on more pause sites for regulating gene expression will be even more complex to study using conventional approaches.

Several chemical roadblocks have been developed to biochemically characterize specific nascent RNA molecules[12,33,36,66,67]. Notably, the *lac* repressor was also employed to specifically block

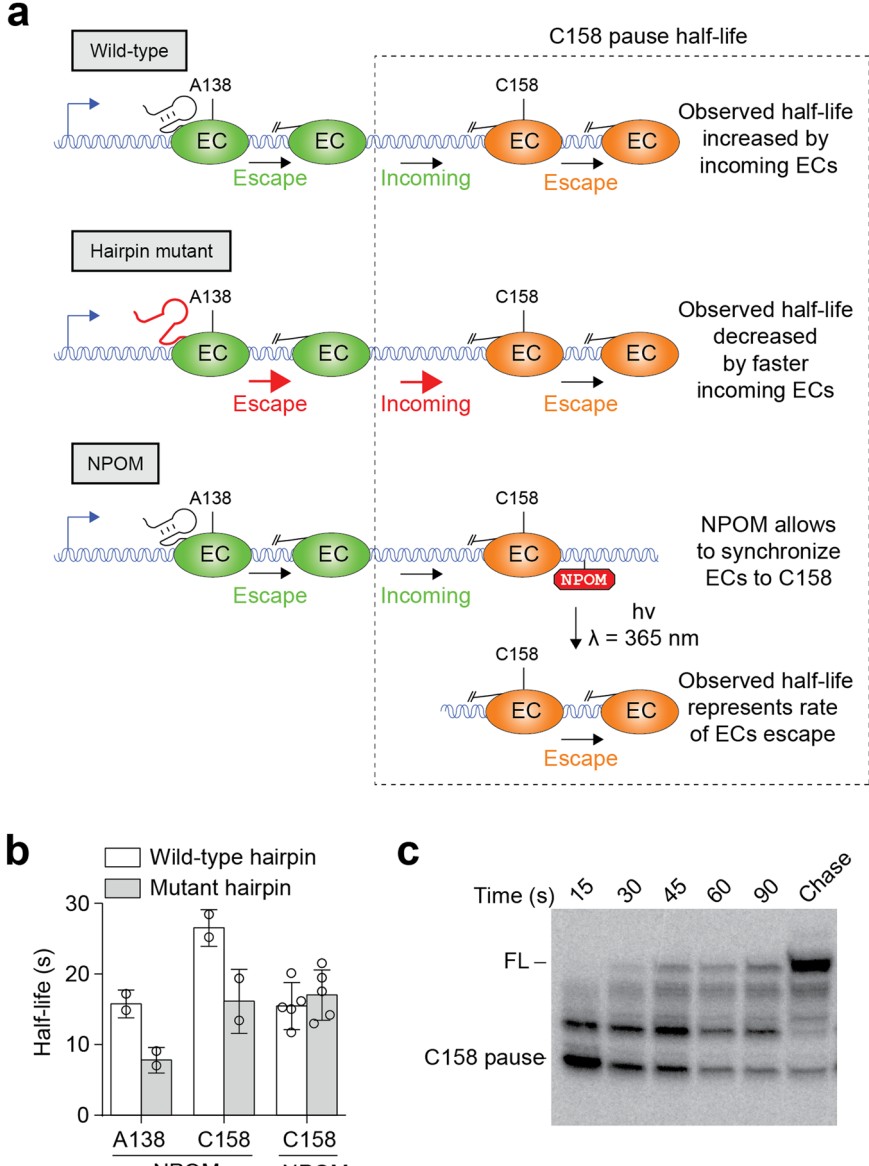

**Fig. 5 Application of NPOM for the study of the C158 pause in the *E. coli thiC* riboswitch. a** The *thiC* riboswitch contains two consecutive transcriptional pauses. Due to its upstream location, ECs escaping the A138 pause may artificially increase the half-life of the C158 pause. In the context of a A138 hairpin mutant, ECs escape the A138 pause at a faster rate, therefore artificially decreasing the C158 pause half-life. Using an NPOM-containing template to synchronize EC at the C158 pause site, the measured C158 pause half-life is expected to strictly represent the rate of pause escape, which should not be affected by the half-life of the A138 pause. **b** Observed half-life values of pauses A138 and C158 in the context of the wild-type and mutant hairpin constructs. Experiments were performed with and without NPOM when monitoring the C158 pause. The average values of independent experiments with standard deviations are shown. **c** Transcription kinetics experiment monitoring the half-life of the C158 pause in the context of the hairpin mutant. ECs were synchronized at the C158 pause site using a NPOM roadblock. NPOM removal was performed in the absence of NTPs. The experiments were performed at least two times and variations were less than 10%.

elongating transcription complexes at specific sites in vivo[67]. Importantly, because such techniques are compatible with multi-subunits RNA polymerases such as bacterial RNAPs, they allow to study correctly folded bacterial nascent transcripts. However, due to the intrinsic properties of these roadblock systems, it is unpractical to remove the roadblock and to release stalled ECs without disrupting nascent RNA-protein and RNA–RNA contacts. The recently obtained dCas9 roadblock provides a significant advance toward this goal as it enables the independent control of multiple ECs through guide RNAs[35]. In contrast, although NPOM roadblocking system described here may only be used to study a single pause at a time, it allows to minimally alter the pause site sequence. This feature is

mostly possible due to the fact that NPOM can be cleaved from dT following UVA exposure, which is a non-invasive technique that is routinely used to control biological processes[42,43,50]. Furthermore, we show here that NPOM may also be used as a roadblock for Taq and Phusion DNAPs.

NPOM-dependent synchronization of ECs at a pause site may allow to directly monitor pause escape and to accurately assess the associated half-life. The use of NPOM in the study of the *his* pause is in agreement with a previous study showing that the rate of pause escape is both influenced by the upstream hairpin and the interaction with NusA[29]. Interestingly, our results revealed that the full-length template showed an increased pause half-life

compared to the minimal length in the absence of NPOM synchronization (Fig. 4b). These results suggest that the increased half-life obtained with the full-length template is probably due to the presence of asynchronous incoming EC populating the *his* pause, a feature that is most likely less occurring in the minimal length template. NPOM-containing templates are complementary to nucleic acids scaffolds as they can allow to block elongating RNAP at virtually any site along a DNA template which thus can be used to study the influence of upstream RNA sequences on pause escape. A recent example of transcriptional pausing was found in the *E. coli thiC* riboswitch where a crucial pause site is located at position C187 in the 5′ UTR[12]. Although it was found in that study that the pause half-life is affected by the riboswitch-ligand interaction, the sequence and structural requirements for transcriptional pausing are still under investigation (Hien, Nadon, and Lafontaine, in preparation) and may require a long RNA stretch upstream of the pause site in the ligand-free form. NPOM-containing templates will be instrumental in characterizing transcriptional pausing mechanisms relying on complex upstream RNA structures such as those found in bacterial riboswitches and ribozymes[1].

In our study, NPOM was also employed to investigate two consecutive pause sites (A138 and C158) of the *thiC* riboswitch[12,28]. We previously observed that mutations introduced in the A138 hairpin affected the half-life of both A138 and C158 pauses[28]. The influence of the A138 hairpin on the C158 pause half-life could not readily be explained by a direct effect from the RNA structure, which therefore prompted us to investigate the cause of the C158 pause half-life variation. Using NPOM-containing templates, our results clearly indicated that the destabilization of the A138 hairpin does not influence the rate of C158 pause escape (Fig. 5b). Because the calculated C158 pause half-life is decreased in the mutant A138 hairpin without NPOM (Fig. 5b), it strongly suggests that asynchronous ECs escaping the A138 pause site affects the determined half-life of the C158 pause (Fig. 5a, upper panel). As a result, when employing conventional approaches not relying on NPOM synchronization, it suggests that the calculated half-life of the C158 pause does not represent the actual pause half-life that strictly depends on the rate of pause escape[28] (Fig. 1b). Due to the prevalence of transcriptional pause sites recently identified[6,64,65], our study suggests that NPOM could be used to characterize the transcriptional regulation of bacterial mRNAs.

Based on our findings, NPOM constitutes a powerful tool to study transcriptional complexes at virtually any locations on a DNA template. Compared to other EC synchronization methods, NPOM is a unique transcriptional roadblock as it can be removed using conditions compatible with cotranscriptional studies. Importantly, in contrast to other approaches relying on conventional DNA templates, the use of NPOM-containing templates allows to monitor specific properties of ECs restarting from a given position along the DNA template. For example, NPOM could be used in a vast array of biochemical assays requiring the use of stable ECs, such as single-molecule fluorescence, RNA-protein cross-linking, and cryo-electron microscopy studies. Furthermore, although stepwise transcription reaction could in principle be used to specifically position RNAP at any site, the loss of material at each step precludes its use to study pause sites located remotely from the TSS (~75 nt). The incorporation of NPOM may be performed by inserting a single point mutation in case no dT is present at the desired location. It is possible that such a single point mutation may alter the half-life or pause efficiency and control experiments are required to avoid such a situation. Alternatively, it is possible to insert NPOM at a naturally occurring dT residue and to perform stepwise transcription to position the RNAP at the downstream pause site to be studied.

Our study also indicates that although NPOM can be used to prevent elongation for a variety of polymerases, NPOM removal does not always allow to resume elongation as observed for the T7 RNAP. Further studies will be required to specifically address the interaction between T7 RNAP and NPOM, which could either result in ECs being unstable or blocked in an inactive conformation.

## Methods

**Simulation of transcriptional pause half-life**. The analysis of transcriptional pausing may be achieved by fitting the fraction of escaping ECs using single or multiple exponential decays[1–3]. It is expected that ECs transiting via a unique transcriptional pause are characterized using a single kinetic constant, which is dictated by the strength of the pause. However, in the case where upstream pause sites are also present, it is likely that multiple kinetic constants are involved, due to asynchronous transcription elongation, which could affect the apparent half-life obtained using single-exponential decay. To demonstrate this aspect, we provide here a theoretical example showing that variations in the rate of incoming EC may affect the apparent rate of downstream pause sites obtained when fitting the data with a single-exponential decay.

For a single transcriptional pause in which there is no incoming RNAPs, the number of RNAPs follows an exponential decay that can be calculated using:

$$N(t) = N_0 e^{-\lambda t} \tag{1}$$

where $N(t)$ is the number of paused RNAP molecules, $N_0$ is the initial population, $\lambda$ is the decay constant (relative to the half-life ($t_{1/2}$), $\lambda = \ln2/t_{1/2}$) and t is time.

The rate of decay is proportional to the population and is given by:

$$\frac{dN}{dt} = -\lambda N \tag{2}$$

In a context where a second transcriptional pause ($P_B$) is located downstream of the first ($P_A$), its decay will also follow an exponential decay.

$$\frac{dN_{BDecay}}{dt} = -\lambda_B N_B \tag{3}$$

However, since RNAPs are escaping $P_A$ and feeding the $P_B$ population, the rate of $P_B$ formation is given by:

$$\frac{dN_{BFormation}}{dt} = \lambda_A N_A \tag{4}$$

Hence, the rate of total variation of $P_B$ is given by:

$$\frac{dN_B}{dt} = \frac{dN_{BFormation}}{dt} + \frac{dN_{BDecay}}{dt} \tag{5}$$

$$\frac{dN_B}{dt} = \lambda_A N_A - \lambda_B N_B \tag{6}$$

An analytical solution to this equation can be given as:

$$N_B = \frac{\lambda_A}{\lambda_B - \lambda_A} N_{A_0} (e^{-\lambda_A t} - e^{-\lambda_B t}) \tag{7}$$

This solution corresponds to the Bateman's equation. Using this formula and simulating two pauses ($P_A$ ($t_{1/2}$ = 0.1, 1, 10 or 30 s) and $P_B$ ($t_{1/2}$ = 5 s), and try to fit the population of $P_B$ with a single exponential decay, we find that, even though the half-life ($t_{1/2}$) of $P_B$ is fixed at 5 s, its apparent value is of 5.0, 5.6, 13.7 and 41.1 s for a $P_A$ half-life of 0.1, 1, 10, and 30 s, respectively. Our simulations are based on the assumption that the efficiency of both pause sites is 100%.

**Calculations of *thiC* C158 pause half-life using a two-step kinetic model**. Using GraphPad Prism, the A138 population ($N(t)$) was first fit with a single-exponential decay to obtain the values of $\lambda_{A138}$, $N_{A138}(0)$ and the Plateau of the function.

$$N_{A138}(t) = (N_{A138}(0) - Plateau) * e^{-\lambda_{A138}*t} \tag{8}$$

To calculate $\lambda_{C158}$, we used Bateman's equation using C158 population ($N_{C158}(t)$), the initial A138 population ($N_{A138}(0)$ - Plateau) and $\lambda_{A138}$, the last two obtained via the previous fitting of A138 population to a single exponential decay. Using a C++ program, the Bateman's equation was solved by the iteration method. The program is using values of $N_{A138}(0)$, $\lambda_{A138}$, and an array of values of $N_{C158}(t)$. After multiple iterative calculations allowing to reach a specified accuracy of $10^{-6}$, the program obtains an array of $\lambda_{C158}$ values which are then averaged and converted to half-live values.

**DNA templates preparation by primer extension and polymerase chain reaction (PCR)**. Primer extensions were performed in Tris-HCl pH 8.8 (20 mM), MgCl$_2$ (3 mM) and KCl (100 mM) when using the EasyTaq DNA polymerase and Tris-HCl pH 8.8 (20 mM), ammonium sulfate (10 mM), KCl (10 mM), Triton X-100 (1%), MgSO$_4$ (2 mM) and BSA (100 μg/mL) when using the Pfu DNA polymerase. The reaction was performed by adding dNTPs (200 μM), the forward template (500 nM), the reverse primer (500 nM), and EasyTaq or Pfu DNA

Polymerase (0.05 U/µL, Transgen Biotech), in a total volume of 600 µL. The temperatures used were: 95 °C for 2 min, 2 cycles (95 °C for 30 s, 58 °C for 30 s, 72 °C for 1 min), 72 °C for 5 min. Polymerase chain reactions were performed like primer extensions, with the addition of *E. coli* genomic DNA (10 µg/mL) in the reaction, which underwent 30 cycles instead of 2.

**Purification of transcriptional complexes using Ni-NTA agarose beads**. 10 µL of Qiagen Ni-NTA Agarose beads washed thrice with transcription buffer were added to the transcription reaction, which was then incubated at 37 °C for 10 min. The beads bound to the transcription complexes via the *E. coli* RNAP histidine tag were then washed twice with 1 mL of solution containing MgCl₂ (10 mM), KCl (700 mM), and Tris (pH 8.0, 40 mM) and twice with 1 mL of transcription buffer.

**GreB cleavage assay**. 5 µL of *pT7A1-tbpA-21-92(85-NPOM)* or *pT7A1-tbpA-21-92(85-biotin)* DNA were mixed with 1 µL (18 pmol) of HIS-tagged core RNAP with 0.25 µL (2.5 µg) σ⁷⁰ in 20 µL of TB50 (40 mM Tris-HCl pH 8.0; 10 mM MgCl₂; 50 mM NaCl). Samples were incubated for 5 min at 37 °C. AUC RNA primer (10 µM) plus ATP and GTP (25 µM) were then added to the reaction. Incubation was continued at 37 °C for 5 min. Samples were transferred to 20 µL Co²⁺ NTA beads with heparin 1.5 mg/ml and shaken 5 min at 22 °C. A volume of 2 µL of [α-³²P] CTP was next added the reaction was incubated for 5 min at 22 °C. Samples were washed twice with 1 mL of TB1000 (40 mM Tris-HCl pH 8.0; 10 mM MgCl₂; 1 M NaCl, 0.003% Igepal-60) and twice with 1 mL of TB100 (40 mM Tris-HCl pH 8.0; 10 mM MgCl₂; 100 mM NaCl, 0.003% Igepal-60). Samples were chased for 2 min by 1 mM NTPs and washed with 1 mL of TB100 4 times. One 10 µL aliquot from each experiment (out of 90 µL) was quenched by 10 µL of SB (1X TBE, 20 mM EDTA; 8 M Urea, 0.025% xylene cyanol, 0.025% bromophenol blue). Samples were split into 50 and 120 µL aliquots each bigger one from each set was transferred into a 96 wells plate in six 20 µL parts. The plate was irradiated at 22 °C in a Strato-Linker at 365 nm, 200 J/cm for 5 min, then samples were transferred back into Eppendorf tubes and spun down, giving a volume of ~50 µL. One 10 µL aliquot was taken from both parts and chased by 1 mM NTPs in the presence of 1.5 mg/mL heparin for 5 min at 22 °C, then quenched by 10 mL of SB. The rest were treated by 300 nM, 1 µM, or 3 µM GreB for 5 min at 37 °C before quenching as above. The samples were heated for 5 min at 100 °C in a dry bath and loaded at 10% (20 × 20 cm) (19:1) polyacrylamide gel with 7 M Urea and 1X TBE. The gel was run 30 min at 50 W. The gel was exposed at phosphor screen overnight.

**Mapping by 3′-OMe-NTP incorporation**. Synchronized halted complexes were prepared as described in E. coli RNAP transcription reactions, but using half the quantity of DNA template, σ⁷⁰ and *E. coli* RNAP. The complexes were purified using G-50 resin. The solution was split in 4 × 10 µL, and 10 µL of 3′-OMe NTP solution (either 3′-OMe-ATP (200 µM), 3′-OMe-UTP (50 µM), 3′-OMe-CTP (200 µM) or 3′-OMe-GTP (100 µM), NTP (100 µM), and heparin (0.45 mg/mL), in transcription buffer) was added. The reaction was allowed to proceed for 10 min at 37 °C, before adding 20 µL of stop solution.

**T7 RNAP transcription reactions**. The T7 RNAP was obtained at the *Plateforme de purification des protéines* (Université de Sherbrooke). The purification procedure was done accordingly to a previous study[4]. Transcription reactions were done in a buffer containing 40 mM Tris-HCl pH 7.5, 15 mM MgCl₂, 5 mM DTT and 2 mM spermidine. Transcription complex formation was carried using 350 nM DNA template, 25 µM CTP/GTP, 0.185 MBq [α-³²P] UTP and 350 nM T7 RNAP (in a total of 10 µL) at 37 °C for 2 min. Transcription elongation was then allowed by incubation with 5 mM ATP, 1.25 mM UTP, 5 mM CTP, 5 mM GTP and 4.5 mg/mL heparin (in a total of 20 µL) at 37 °C for 10 min. Reactions were stopped using at least twice the volume of stop solution.

**GreB factor**. For purification of the GreB protein Bl21 (DE3) cells carrying pET45 GreB- 6XHIS (cloned between XhoI and NcoI) plasmid were grown from a single colony overnight at 37 °C with shaking in 10 mL of LB broth and 0.1 mg/ml carbenicillin. Next day 2 L of LB broth were inoculated with overnight culture and grown at 37 °C with shaking till OD600 = 0.5. GreB expression was induced by addition of 1 mM IPTG for 2 h and cells were harvested 10 min at 5000 g. Resulting cell paste was re-suspended in 40 mL of the lysis buffer (GLB) [50 mM Tris-HCl pH 7.0, 1.2 M NaCl, 5% glycerol, 1 mM 2-mercaptoethanol (ME)] with Complete EDTA-free protease-inhibitor cocktail (Roche), 0.1% Tween 20 and 1 mg per milliliter lysozyme. The resulting mix was incubated 1 h at +4 °C with stirring and sonicated at maximum output for 10 min on ice to complete cell lysis. The lysate was cleared by centrifugation (27,000 × g, 30 min at +4 °C). The cleared lysate was mixed with 10 mL of Co²⁺–NTA agarose (Invitrogen) slurry in GLB buffer and incubated with rotation for 30 min at +4 °C. The slurry was poured into a single-use gravity-flow column and drained. The column was washed with 100 mL of GLB buffer and 50 mL of the same buffer with 10 mM imidazole. Elution was carried out with 50 mL of GLB buffer with 500 mM imidazole. Eluate was concentrated to approximately 3 ml using an Amicon Ultra-15 3000 MW filter-device and loaded onto a HiLoad Superdex 75 16/60 column (GE Healthcare) using an AKTA Purifier system (GE Healthcare) at 0.5 ml per minute. The column was washed with high salt GF buffer [10 mM Tris-HCl pH 7.9, 1 M NaCl, 2 mM dithiothreitol (DTT)].

The resulting peak of GreB protein was concentrated as above, diluted twice with 100% glycerol, flash-frozen in liquid nitrogen, and kept at −80 °C for storage.

**E. coli RNAP transcription reactions**. The *E. coli* RNAP, *E. coli* σ⁷⁰ and *E. coli* NusA factors were obtained at the *Plateforme de purification des protéines* (Université de Sherbrooke). The purification procedures were done accordingly to previous studies[68,69]. Transcription reactions were done in a buffer containing Tris-HCl pH 8.0 (20 mM), MgCl₂ (20 mM), NaCl (20 mM) and EDTA pH 8.0 (100 mM). Transcription complex formation was performed using the appropriate DNA template (400 nM), σ⁷⁰ (400 nM), and *E. coli* RNAP (200 nM) in a total of 10 µL at 37 °C for 5 min. Transcription initiation was then allowed by adding a combination (described in Supplementary Table 2) of initiator di- or trinucleotide (10 µM), a subset of two nucleoside triphosphates (2.5 µM), and a radiolabeled nucleoside triphosphate (0.3 µCi/µL), letting the reaction proceed for 8 min at 37 °C, yielding synchronized halted complexes. Transcription elongation was then performed by adding NTPs (1 mM), heparin (0.45 mg/mL) and, when present, NusA (50 nM) in a total of 20 µL. The mixture was then left to react at 37 °C for 5 min, before adding twice its volume of stop solution (95% formamide, 20 mM EDTA (pH 8.0), and 0.4% SDS, bromophenol blue and xylene cyanol). The DNA templates used in this study are shown in Supplementary Table 3 and the required DNA oligonucleotides are presented in Supplementary Table 4. Oligonucleotides bearing an NPOM-bearing nucleotide were ordered from IDT (4 nmol, standard desalting).

**Purification of transcriptional complexes G-50 resin**. Purification columns were prepared by adding 500 µL of G-50 slurry (2.13 g illustra™ Sephadex™ G-50 Fine DNA Grade resin, Tris (pH 8.0, 10 mM), and EDTA (1 mM, pH 8.0) in a total volume of 40 mL, kept at least 3 h at rt) to an empty Micro Bio-Spin™ Column (Bio-Rad). The column was centrifuged at 600 × g for 1 min and washed with 100 µL of transcription buffer (centrifuge at 600 × g for 1 min) before adding 20 µL or more of the transcription reaction (using transcription buffer if the volume was too small) to the column. Elution was done by centrifuging at 600 × g for 1 min.

**NPOM cleavage by UVA irradiation**. Transcription reactions were performed using NPOM-containing DNA templates to obtain roadblocked ECs. These complexes (50 µL reaction) were irradiated using a handheld UV lamp (Fisher Scientific) set at 365 nm (8 W). Roadblocked ECs were put in a 96 wells plate and irradiation was performed by directly positioning the lamp on the top of the plate at room temperature for 8 min. In such as setup, the lamp is positioned < 3 cm from the irradiated samples and where indicated, different exposure times were used. The UV tube used in these assays corresponds to the model UVP 34000601 (Fisher Scientific).

**Transcription kinetics experiments**. Synchronized complexes halted either after a first step of transcription (NPOM-less templates) or at the position -1 from NPOM (NPOM templates) were prepared as described in E. coli RNAP transcription reactions. The volume of reaction mixtures was increased to 40 µL using 1X transcription buffer, and the complexes were then purified using either G-50 resin (NPOM-less templates) or Ni-NTA agarose beads (NPOM templates). When applicable, NPOM was cleaved following NPOM cleavage by UVA irradiation. Transcription elongation was restarted by adding NTPs (150 µM ATP, UTP and CTP, and 10 µM GTP for *his* templates, 150 µM UTP, CTP and GTP, and 10 µM ATP for *his* G103A and *his* NPOM templates, or 25 µM for *thiC* templates) and heparin (0,45 mg/mL) in a total volume of 80 µL. Aliquots of 4 µL were taken at the indicated time intervals and added to 8 µL of stop solution. When Ni-NTA agarose beads were used, the beads were re-suspended quickly before each aliquot was taken. After the last time point, 1 mM NTP were added and transcription reactions were chased for 5 min.

**Roadblocking and restarting DNA polymerization by the Taq and Phusion DNAPs**. The 3257 JG primer (40 pmol) was radiolabeled using the T4 polynucleotide kinase (New England Biolabs) and [γ-³²P]-ATP, then purified using a G-50 column on the reaction, the volume of which has been raised to 50 µL prior to purification. The radiolabeled primer (4 pmol) was then added to a reaction mixture containing *pLacUV5-tbpA(+5)-92* or *pLacUV5-tbpA(+5)-92(85-NPOM)* DNA template (1 pmol), dNTPs (200 µM) and EasyTaq or Pfu DNA Polymerase (0.05 U/µL, Transgen Biotech), in 100 µL of the appropriate buffer. The reaction mixtures were subjected to a temperature of 95 °C for 2 min 30 s, 58 °C for 5 min and 72 °C for 5 min. Where relevant, NPOM was then cleaved following NPOM cleavage by UVA irradiation. The reactions were stopped by adding the stop solution (200 µL).

**Reporting summary**. Further information on research design is available in the Nature Research Reporting Summary linked to this article.

## Data availability
The source data underlying Figs. 4 and 5 are provided as a source data file. Other data are available in the Supplementary Information and from the corresponding author upon reasonable request.

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

## Acknowledgements

We thank members of the Lafontaine laboratory and Alain Lavigueur for discussion. This work was supported by grants from the Canadian Institutes of Health Research, Natural Sciences and Engineering Research Council of Canada, the Blavatnik Family Foundation, and the Howard Hughes Medical Institute.

## Author contributions

Conceived, designed, and performed experiments: J.-F.N. and V.E. Analyzed the data: J.-F.N., V.E., E.C., M.R.S., A.S.V., E.N., and D.A.L. Participated in writing the manuscript: J.-F.N., V.E., E.N., and D.A.L.

## Competing interests

The authors declare no competing interests.
