## [Peer Review File · Communications Biology]

Reviewers' comments:

Reviewer #1 (Remarks to the Author):

Nadon et al evaluated the suitability of NPOM as a photocleavable roadblock for bacterial RNA polymerase (RNAP). Stopping RNAP in a desired location while preserving the integrity of the elongation complex and the catalytic activity is a surprisingly challenging task. Several classic techniques such as stepwise walking of RNAP with subsets of nucleotides or using DNA binding proteins as roadblocks have been known for some time. However, walking is typically limited to relatively short distances due to significant losses of RNAP at each extension-wash step. DNA binding proteins require a relatively long recognition sequence, are not 100% effective, difficult to remove, and the RNAP tends to backtrack when encountering a strong roadblock. Out of many available natural and unnatural modifications to the oligonucleotide sequences some are expected to halt RNAP, yet most do not permit reactivation of the stalled RNAP. Nadon et al demonstrated that NPOM modification not only efficiently halts RNAP without causing backtracking but can also be efficiently removed by a relatively short illumination with a moderately intense UV light. It is an extremely valuable methodological discovery. Furthermore, Nadon et al presents several case studies demonstrating the usability of the NPOM roadblock and furthering our understanding of transcriptional pausing. I have only minor comments that are detailed below.

1. I did not find any information about the oligonucleotide synthesis. NPOM is commercially available as phosphoramidite and as an internal modification of custom-synthesized oligonucleotides. Were oligos synthesized in house or ordered from a company? What was the purification method, scale?
2. Changing the pause escape nucleotide (G103A substitution) is not a generally recommended approach for studying kinetic properties of a pause. Nadon et al reported that the pause half-life was not affected by the substitution, but that was arguably a lucky coincidence and not something generally expected to be the case. Also, what about the pause efficiency, was it also unchanged? It would have been better to stall RNAP shortly upstream of the pause site. I do not think that the authors need to perform an additional experiment by stalling RNAP upstream of the pause because such experiment would be redundant with ML template experiment. However, I suggest that the authors caution readers against changing the pause escape nucleotide when studying the pause escape kinetics.
3. The authors provide statements about the reproducibility of the experiments and the number of repeats in the reporting summary. Please check the editorial policy whether those statements should also be included in the manuscript, figure legends or both.
4. Fig. 2A, "NPOM" label should be gone after illumination with UV light, only T should remain.
5. Fig. 4 legend: "various constructions" or "various constructs"?
6. Fig. 4 panel A. "NPOM starting position" is arguably a misleading term. Would it be better to write "RNA 3' end in NPOM halted complex" or something along those lines? It would also be beneficial to present a more elaborate schematics of the entire experiment depicting a relevant part of the DNA template as a supplementary figure: RNAP stops before NPOM > washed > NPOM is removed > nucleotides added to measure the escape kinetics.
7. Rate equations in the supplementary material assume that the efficiency of both pause sites is 100%. That is OK for a proof-of-principle simulation, but real-life pauses are typically off-pathway states with efficiency less than 100%. Hence it is important to state that the efficiency was assumed to be 100% for the sake of simplicity whereas the real-life scenarios will be more complex.
8. The removal of NPOM by illumination is a relatively slow process. While some reactivation assays can be done in the presence of NTPs, for kinetic measurements it is imperative to wash

away NTPs, reactivate and then add NTPs back. I acknowledge that was how the experiments were performed, but perhaps it would be beneficial to explicitly state for each experiment (and figure legend) whether reactivation was done in the presence of NTPs or in the absence of NTPs.

Reviewer #2 (Remarks to the Author):

In the manuscript by Nadon et al., the authors present a significant advance in the field, wherein they engineered a photolabile modification, NPOM, to study pausing of transcription elongation complexes. The authors demonstrate clearly that NPOM creates a stable "roadblock" for bacterial RNA polymerase (RNAP), which then is releasable upon excitation at 365 nm. They further show that this roadblock does not cause backtracking, as demonstrated by lack of cleavage product in the presence of transcription factor GreB. Lastly, the authors use this technology to study the thiC riboswitch, which contains multiple pause sites, the latter of which cannot be described by single-exponential decay functions. Overall, the manuscript presents a clear narrative that is of scholarly character and will be of great interest to the transcription field.

One concern with the manuscript is, however, the oversimplification of the field's struggle to mathematically model pausing. As the authors demonstrate in Figure 1, fitting with (not "to") a single-exponential function cannot resolve the kinetics of pausing since there is a delay associated with RNAP reaching the pause (independent of whether there are upstream pauses or not). However, there have been significant advances made in kinetic simulations and fitting since this issue was first noted (Landick et. al. 1996). The authors should consider commenting on global fitting strategies that have since allowed for resolution of more complicated enzymatic schemes, including multiple pausing events where sequences pauses are best represented by a coupled kinetic scheme and equation system. The combined use of the NPOM photolabile roadblock in vitro in concert with the computational advances of current fitting strategies in silico will indeed be exciting for the field of transcription kinetics, by resolving this complication experimentally and computationally, respectively.

With clarification of the fitting advances in the field discussed above, as well as further discussion and exposition of a few major/minor comments listed below, this manuscript will be of great quality and interest to the transcription field.

Major Comments:

1. The problem presented with kinetic modeling of multiple pause sites is a bit outdated (Landick paper is from 1996!). In the time of lower computational power, this modeling was constrained to model-independent fitting, often with a single-exponential decay curve that ignored the initial rise part. With greater computational power, current model-dependent fitting strategies allow for constraining multiple kinetic parameters simultaneously (Ray-Soni et al. PNAS, 2017; Ken Johnson The Enzymes 1992; Dangerfield et al. iScience, 2020, Nakajima et al. Journal of Theoretical Biology, 2020, and many others) using numerical integration or, even better, analytical matrix algebra. The authors ought to include both model-independent and model-dependent simulations in their argument and compare those results in the absence of NPOM with the experimental solution of using NPOM at a specific downstream pause site. I.e., what happens to the thiC riboswitch k_{app} values when the data are fit with the analytical solution for a two-step kinetic pathway? In this way, the authors could present their technology as an alternative or additional – experimental – means by which model-dependent simulations could be experimentally probed.
2. More description of the differences between the site-specific dCas9 and NPOM release strategies (the latter of which can only be used at a single pause at a time, in contrast to the former) in the Introduction and/or Conclusions would be useful to those in the field weighing the pros/cons between these two techniques.
3. Important experimental information is missing throughout, and especially in Figures 2 and 3, such as the number of replicates carried out and reproducibility of results (error, standard deviation, etc.).
4. Figure 2 – nucleotide starvation of one NTP is another method often used by the field to halt polymerases. Can the authors comment on this as an additional positive control?

5. Figure 3 – removal of NPOM is relatively slow in comparison to the rate of transcription kinetics and appears to be variable between single RNA molecules (e.g., 60% RT vs 40% photo-blocked at 4 minutes), in bulk – can the authors comment on why this is the case, and what limitations this might bring to the experimental interpretation?

6. In the Supplemental Methods section on “simulation of transcriptional pause half-life”, it is unclear why the authors are attempting to fit their simulations with a single-exponential decay, when they clearly can describe their data with an analytical solution describing a two-step kinetic model. It seems fairly obvious that the values derived would not be the same in this case, and since the authors are knowingly fitting the data with a function that they know does not describe them well, they should consider switching to a more appropriate double-exponential.

Minor Comments:

1. Can the authors comment on why they see 73% of restart activity in 3C, but only 48% in 3D?

2. Can the authors estimate the extent of pyrimidine dimer formation after 8 minutes of UVA exposure?

3. It would be useful to discuss why T7 RNAP could not be restarted, but E. coli transcription could.

4. In lines 131-133, the authors state that transcriptional readthrough is not observed, however, in Figure 2B there appear to be faint bands at the full-length size.

5. In line 172 it is at first not clear that transcription was performed subsequent (not in parallel) to UVA exposure – this should be clarified.

6. On line 235, the authors say they monitor the “kinetics of transcription elongation”, however, there is no information here describing transcription elongation – only transcriptional pausing. This language should be clarified.

7. On line 414 it is debatable whether the claim that NPOM can be “selectively” removed since only a single NPOM can be removed so that removal of this NPOM is not selective over any others.

“Selectivity” implies a comparative selection of one site over another, which is not the case here.

8. Figure 4 – in the text it is difficult to pick apart what the purpose of the “full length” vs.

“minimal length” constructs are. The authors mention Figure 4a in line 229, but do not describe the minimal length transcript until line 265. If the minimal length construct were described earlier in the text, it would increase the readability of this section.

9. Many of the Figure cartoons seem a bit primitive (in the era of Biorender and Photoshop); in particular, Figures 1A, 3A, 5A would benefit from a facelift to a more sophisticated representation of the DNA, RNAP and RNA.

Typographical errors:

10. On line 52, the authors say “the half-life of pause site is directly affected”. This should read as “the half-life of a pause site...”.

11. On line 166 it should read: “taking place within”.

12. On lines 174/175 should read: “UVA illumination...leads”.

13. On line 183 it should read: “In addition to efficiently blocking...”.

14. On line 279 it may be better to specify “specifically slows pause escape by”.

15. On line 280. Delete the first use of “previous”.

16. On line 420, a hyphen can be added to the word “cryo-electron”.

17. NPOM’s full UIPAC name should be given upon first introduction both in the Abstract and main text.

18. It should always be “fitting a function to data” or “fitting data with a function” rather than the other way around.

Reviewer #3 (Remarks to the Author):

Nadon et al. report use of a photolabile thymidine analog (6-nitropiperonyloxymethyl-thymidine) to roadblock RNA polymerase and then release it for kinetic analyses of transcriptional pausing by photogeneration of the thymidine base. This method of halting and releasing RNA polymerase offers distinct advantages over other available methods to halt RNA polymerase and its use for this purpose has not been reported previously. The authors validate the method by performing kinetic analyses on the well-characterized E. coli his leader pause site, replicating several well-known features of this site but also finding evidence that the nature of the initial halt site upstream from

a pause or transcription of the region between an initial halt site and the pause site can affect the apparent dwell time of RNAP at the pause site. The authors also use the method to test the previous finding that altering a pause site upstream from a pause in the *thiC* leader region can affect the apparent dwell time of RNAP at the downstream pause site, but that this effect disappears when RNAP is restarted at the second pause site using the NPOM roadblock technique. The results will be of broad interest to researchers in the field of transcription, both prokaryotic and eukaryotic, and merit publication. Although new insights into transcriptional regulatory mechanisms from the present study are limited, the significance of work in establishing the utility of a robust new method is high. Nonetheless, there are a few aspects of the work that could be improved either by increasing the clarity of the manuscript or adding information. These suggestions are listed below.

1. The NPOM method is the most exciting and impactful aspect of this work, but it is surprisingly underdescribed. I could not find a description of the temperature at which the photorestitution reactions were conducted or an estimate of the photon flux used. These basic aspects of the method should be highlighted and systematically described. Even if the authors are unable to measure photon flux, a detailed description of the UV source used should be included (what model of UV lamp was used and what fraction of the 8W power is actually directed to the sample?). Did the authors explore use of different temperatures or UV doses to investigate whether an optimum exists for high levels of T generation with lower levels of transcription complex inactivation? What is the mechanism of EC inactivation during the photolysis reaction? Is it due to dissociation of ECs during the photolysis reaction or some other mechanism? Does addition of GreA/B release a greater fraction of ECs after the photolysis treatment, indicating arrest backtracking as the mechanism? At a minimum, a complete description of reaction conditions and a description of whether complex inactivation increases with time of UV exposure should be included.
2. The authors report that reducing the length of DNA upstream of the *his* pause causes a reduction in the apparent half life, but they do not report the rate at which RNAP transcribes from the initial halt site to the pause site for the two templates studied. At a minimum, the authors should show the portions of the gels with the halted complexes in the figures. It seems possible the difference could result from differences in the rate of escape from the initially halted complexes rather than the length of DNA transcribed. This doesn't directly bear on the utility of the NPOM method, but is important to ensure the authors aren't mis-describing their results.
3. The failure to see a NusA effect on the *his* pause using the NPOM method is confusing and the explanation offered does not accord with prior demonstrations that NusA enhances pausing at the *his* pause by ECs halted just 1 nt before the pause site (e.g., Guo et al., 2018 Mol. Cell 69:816). Perhaps some sort of mixing experiment could be performed to test if the NusA is active on other complexes in the same solution. Given prior findings, it seems possible that some sort of experimental artifact has airen in this experiment.
4. Although the authors mention how halting ECs with NPOM allows tests of the effects of upstream events on pausing, the descriptions focus primarily on pause kinetics and don't highlight other possible applications such as preparing ECs for structural studies in arbitrary sequence contexts using RNA synthesis rather than direct reconstitution. I think highlighting some other applications as well as making the point that the method allows a formal distinction between the properties of ECs actively transcribing though a template position vs. restarting after being halted artificially at a template position could raise the impact of the manuscript. The authors should consider adding a section to the discussion making these points.

Minor issues

1. The description of current methods for measuring pause dwell times as being biased seems misleading. An extensive literature already recognizes the difference between apparent pause dwell times and the intrinsic kinetics of pausing. Although the authors at times use more appropriate descriptions of "apparent half-lives" rather than suggesting that unsynchronized methods are being used to measure actual dwell times, they should be careful to always make this distinction (e.g., line 57 should refer to apparent half life not half-life) and not to suggest that the

distinction between apparent dwell times and actual rates of pause escape is not already well established in the transcription literature. Indeed in a paper the authors should cite (e.g. at line 53), Theissen et al., 1990 (Anal. Biochem. 189:254) thoroughly describe this distinction as well as method to quantify pause strengths independent of asynchrony effects. In addition, true pause escape rates can be elucidated by kinetic modeling, and that approach also has been used extensively. My recommendation would be to avoid the use of "bias" as a descriptor and simply make that point that asynchronous transcription is well known to cause apparent dwell times to be longer than actual pause escape rates – a problem the authors method can deconvolute.

2. The penultimate sentence of the abstract is confusing. It's unclear if the authors mean the methods are contrasting or the results are contrasting since the sentence suggests a method is in contrast with a result. Consider revising this sentence.

3. In the introduction, the authors should consider adhering to scientific convention by describing already established (published) findings in the present tense rather than in the past tense.

4. li60 It is misleading to suggest that conventional methods lead to biased conclusions, especially without citing an example, as it implies this distinction between true pause half-life and apparent pause half-life has not previously been appreciated, which is not the case. It would be more accurate to say unsynchronized transcription through a pause site can cause apparent dwell times to be greater than an accurate measurement of pause escape rate would reveal.

5. li72 and other locations (e.g. li377, 385, 386) – "...enzyme...allows to block...elongation." There are multiple uses of this incorrect grammatical construction, which combines an active verb with an infinitive, in the manuscript. This example could be reworded more simply as "...enzyme...can block...elongation." or "...enzyme...can be used to block...elongation." The other cases should also be reworded to avoid confusing readers about the authors' intended meaning.

6. li89 – delete "Clearly, " At the start of the sentence.

7. li140 – The authors suggest that NPOM prevents transcription elongation by inhibiting binding of ATP to NPOM-T in the template DNA strand. It is also possible that NPOM-T fails to translocate into the active site. Failure to translocate would cause RNAP to stall 1-bp earlier on the DNA template than if NPOM-T prevents ATP binding after translocation but would give the same RNA 3' end in the halted transcription complex. It would be ideal if the authors could distinguish and report on these two possibilities, for example by using exoIII footprinting or by testing for the sensitivity of the halted complex to pyrophosphate. At a minimum, the authors should mention both possibilities.

8. li166 – The authors show that NPOM-halted transcription complexes are less prone to backtracking than streptavidin-halted transcription complexes. It is unclear, however, if NPOM-T inhibits backtracking or if streptavidin promotes backtracking. The authors could test whether NPOM-T actively inhibits backtracking by halting RNAP at a site known to be susceptible to backtracking and using Gre cleavage or exoIII footprinting to assess the susceptibility to backtracking of complex halted by NPOM-T vs. those halted by NTP-deprivation (eg, because the reaction mix lacks ATP). Although this experiment is not required for publication of the authors work, it would enhance their paper. At a minimum, the authors should mention that their result could be explained either because NPOM-T inhibits backtracking or because streptavidin promotes backtracking.

9. li200 – The authors report that a fraction of ECs are not in an active conformation after photolysis of NPOM. As described in major point 1 above, the authors should attempt to determine an explanation for this inactivation. At a minimum, the authors should offer possible explanations for the inactivation.

10. li211 – The authors report that T7 RNAP halted by NPOM does not resume transcription after NPOM photolysis. T7 RNAP is known to form relatively unstable ECs. Is the failure to resume transcription due to T7 RNAP dissociating from DNA during the photolysis reaction or to some other form of inactivation? If T7 RNAP is halted at the same site by simple NTP deprivation, is it

equally unable to restart, or does the photolysis reaction inactivate the halted T7 RNAP?

11. li306 – should be “apparent half-life”.

12. li374 – “...NPOM may also be used as a roadblock for T7 RNAP...”. This statement is misleading because the authors have not established that T7 RNAP remains on DNA after encountering the NPOM roadblock.

13. li391 delete “s” from “investigations”.

14. Methods section – The authors do not describe the sources of NPOM oligos, RNAP, NusA, Gre factors, or other enzymes used in the studies. This information must be included to allow enable reproduction of the authors work. For a paper in which an important new method is being reported, I found the Methods section surprisingly sparse on information. The methods for NPOM cleavage (li452) should provide all relevant information.

Notes:

1. All corrections performed in the manuscript are colored in red.
2. Andrey Vasenko and Mikhail Samatov have been added as authors since they has participated in elaborating new data.

****Referee 1****

Nadon et al evaluated the suitability of NPOM as a photocleavable roadblock for bacterial RNA polymerase (RNAP). Stopping RNAP in a desired location while preserving the integrity of the elongation complex and the catalytic activity is a surprisingly challenging task. Several classic techniques such as stepwise walking of RNAP with subsets of nucleotides or using DNA binding proteins as roadblocks have been known for some time. However, walking is typically limited to relatively short distances due to significant losses of RNAP at each extension-wash step. DNA binding proteins require a relatively long recognition sequence, are not 100% effective, difficult to remove, and the RNAP tends to backtrack when encountering a strong roadblock. Out of many available natural and unnatural modifications to the oligonucleotide sequences some are expected to halt RNAP, yet most do not permit reactivation of the stalled RNAP. Nadon et al demonstrated that NPOM modification not only efficiently halts RNAP without causing backtracking but can also be efficiently removed by a relatively short illumination with a moderately intense UV light. It is an extremely valuable methodological discovery. Furthermore, Nadon et al presents several case studies demonstrating the usability of the NPOM roadblock and furthering our understanding of transcriptional pausing. I have only minor comments that are detailed below.

Ref1, Major Concern 1.

Referee 1: I did not find any information about the oligonucleotide synthesis. NPOM is commercially available as phosphoramidite and as an internal modification of custom-synthesized oligonucleotides. Were oligos synthesized in house or ordered from a company? What was the purification method, scale?

Response by Authors: We thank the Referee for this comment. To specify the source of NPOM, we have added this in the Methods section (line 484): "Oligonucleotides bearing an NPOM-bearing nucleotide were ordered from IDT (4 nmol, standard desalting)."

Ref1, Major Concern 2.

Referee 1: Changing the pause escape nucleotide (G103A substitution) is not a generally recommended approach for studying kinetic properties of a pause. Nadon et al reported that the pause half-life was not affected by the substitution, but that was arguably a lucky coincidence and not something generally expected to be the case. Also, what about the pause efficiency, was it also unchanged? It would have been better to stall RNAP shortly upstream of the pause site. I do not think that the authors need to perform an additional experiment by stalling RNAP upstream of the pause because such experiment would be redundant with ML template experiment. However, I suggest that the authors caution readers against changing the pause escape nucleotide when studying the pause escape kinetics.

Response by Authors: As indicated by the Referee, it is possible that substituting the residue at the pause site may change the pause half-life. As requested, we have added this in the Discussion (line 453): "It is possible that such a single point mutation may alter the half-life or pause efficiency and control experiments are required to avoid such a situation."

Ref1, Major Concern 3.

Referee 1: The authors provide statements about the reproducibility of the experiments and the number of repeats in the reporting summary. Please check the editorial policy whether those statements should also be included in the manuscript, figure legends or both.

Response by Authors:

We thank the Referee for this and we have added the information in the figure legends.

Ref1, Major Concern 4.

Referee 1: Fig. 2A, “NPOM” label should be gone after illumination with UV light, only T should remain.

Response by Authors: This has been corrected.

Ref1, Major Concern 5.

Referee 1: Fig. 4 legend: “various constructions” or “various constructs”?

Response by Authors: "Various constructions" has been changed to "various constructs".

Ref1, Major Concern 6.

Referee 1: Fig. 4 panel A. “NPOM starting position” is arguably a misleading term. Would it be better to write “RNA 3’ end in NPOM halted complex” or something along those lines? It would also be beneficial to present a more elaborate schematics of the entire experiment depicting a relevant part of the DNA template as a supplementary figure: RNAP stops before NPOM > washed > NPOM is removed > nucleotides added to measure the escape kinetics.

Response by Authors: As suggested by the Referee, we have added "RNA 3' end in NPOM halted complex" in the Fig. 4A. We have also added a new Supplementary Fig. 4 describing in more details the experiments, as suggested.

Ref1, Major Concern 7.

Referee 1: Rate equations in the supplementary material assume that the efficiency of both pause sites is 100%. That is OK for a proof-of-principle simulation, but real-life pauses are typically off-pathway states with efficiency less than 100%. Hence it is important to state that the efficiency was assumed to be 100% for the sake of simplicity whereas the real-life scenarios will be more complex.

Response by Authors: We thank the Referee for this comment. We have added this sentence in the Supplementary Methods section (line 142): "Our simulations are based on the assumption that the efficiency of both pause sites is 100%"

Ref1, Major Concern 8.

Referee 1: The removal of NPOM by illumination is a relatively slow process. While some reactivation assays can be done in the presence of NTPs, for kinetic measurements it is imperative to wash away NTPs, reactivate and then add NTPs back. I acknowledge that was how the experiments were performed, but perhaps it would be beneficial to explicitly state for each experiment (and figure legend) whether reactivation was done in the presence of NTPs or in the absence of NTPs.

Response by Authors: To provide more details about how experiments were done, we have added the sentence "NPOM removal was performed in the presence of NTPs" to legends of Fig. 3C, 3D, 3E, 3F and Supplementary Fig. 5A-C. We also have added the sentence "NPOM removal was performed in the absence of NTPs" to legends of Fig. 4D, 4E and 5C and Supplementary Fig. 6.

****Referee 2****

In the manuscript by Nadon et al., the authors present a significant advance in the field, wherein they engineered a photolabile modification, NPOM, to study pausing of transcription elongation complexes. The authors demonstrate clearly that NPOM creates a stable “roadblock” for bacterial RNA polymerase (RNAP), which then is releasable upon excitation at 365 nm. They further show that this roadblock does not cause backtracking, as demonstrated by lack of cleavage product in the presence of transcription factor GreB. Lastly, the authors use this technology to study the thiC riboswitch, which contains multiple pause sites, the latter of which cannot be described by single-exponential decay functions. Overall, the manuscript presents a clear narrative that is of scholarly character and will be of great interest to the transcription field.

One concern with the manuscript is, however, the oversimplification of the field’s struggle to mathematically model pausing. As the authors demonstrate in Figure 1, fitting with (not “to”) a single-exponential function cannot resolve the kinetics of pausing since there is a delay associated with RNAP reaching the pause (independent of whether there are upstream pauses or not). However, there have been significant advances made in kinetic simulations and fitting since this issue was first noted (Landick et al. 1996). The authors should consider commenting on global fitting strategies that have since allowed for resolution of more complicated enzymatic schemes, including multiple pausing events where sequences pauses are best represented by a coupled kinetic scheme and equation system. The combined use of the NPOM photolabile roadblock in vitro in concert with the computational advances of current fitting strategies in silico will indeed be exciting for the field of transcription kinetics, by resolving this complication experimentally and computationally, respectively.

With clarification of the fitting advances in the field discussed above, as well as further discussion and exposition of a few major/minor comments listed below, this manuscript will be of great quality and interest to the transcription field.

Ref2, Major Concern 1.

Referee 2: The problem presented with kinetic modeling of multiple pause sites is a bit outdated (Landick paper is from 1996!). In the time of lower computational power, this modeling was constrained to model-independent fitting, often with a single-exponential decay curve that ignored the initial rise part. With greater computational power, current model-dependent fitting strategies allow for constraining multiple kinetic parameters simultaneously (Ray-Soni et al. PNAS, 2017; Ken Johnson The Enzymes 1992; Dangerfield et al. iScience, 2020, Nakajima et al. Journal of Theoretical Biology, 2020, and many others) using numerical integration or, even better, analytical matrix algebra. The authors ought to include both model-independent and model-dependent simulations in their argument and compare those results in the absence of NPOM with the experimental solution of using NPOM at a specific downstream pause site. I.e., what happens to the thiC riboswitch kapp values when the data are fit with the analytical solution for a two-step kinetic pathway? In this way, the authors could present their technology as an alternative or additional – experimental – means by which model-dependent simulations could be experimentally probed.

Response by Authors: We thank the Referee for allowing us to better position our study versus the analysis of two-step kinetic pathways. Specifically, concerning the relevant question "What happens to the thiC riboswitch Kapp values when the data are fit with the analytical solution for a two-step kinetic pathway?", we have used a two-step kinetic model to obtain evidence that the A138 hairpin mutant does not influence the half-life of the C158 pause. To do so, we have received help from Prof. Andrey Vasenko and his student Mikhail Samatov (HSE University, Moscow). They have written a program in C++ that solves the Bateman equation by the iteration method. This program is using values of $N_{A138}(0)$ and λ_{A138} and an array of values of $N_{C158}(t)$. After multiple iterative calculations allowing to reach a specified accuracy of 10^{-6} , the program obtains an array of λ_{C158} values which are then averaged and converted to half-live values. The results were added in Supplementary Table 1 and have been introduced in the

Results section. A description of this analysis is also included in the Supplementary Methods under the section "Calculations of *thiC* C158 pause half-life using a two-step kinetic model". We also have added more recent references about the study of multiple pause sites in the main text (line 57).

Ref2, Major Concern 2.

Referee 2: More description of the differences between the site-specific dCas9 and NPOM release strategies (the latter of which can only be used at a single pause at a time, in contrast to the former) in the Introduction and/or Conclusions would be useful to those in the field weighing the pros/cons between these two techniques.

Response by Authors: We agree with the Referee that the dCas9 roadblock is highly useful as it may be used to simultaneously monitor multiple pause sites, in contrast to NPOM. We have therefore rephrased a section of the Discussion section to make this clear (line 397): "The recently obtained dCas9 roadblock provides a significant advance toward this goal as it enables the independent control of multiple ECs through guide RNAs. In contrast, although NPOM roadblocking system described here may only be used to study a single pause at a time, it allows to minimally alter the pause site sequence".

Ref2, Major Concern 3.

Referee 2: Important experimental information is missing throughout, and especially in Figures 2 and 3, such as the number of replicates carried out and reproducibility of results (error, standard deviation, etc.).

Response by Authors: This information has been added in the legends of each figure. For instance, gel experiments were at least performed two times and showed less than 10 % variations. The average and standard deviations are shown for the beta-galactosidase data in the respective figures.

Ref2, Major Concern 4.

Referee 2: Figure 2 – nucleotide starvation of one NTP is another method often used by the field to halt polymerases. Can the authors comment on this as an additional positive control?

Response by Authors: Nucleotide starvation or stepwise transcription, is a technique to allow the production of halted ECs. As correctly pointed out by the Referee, stepwise transcription can be used on its own to generate halted complexes at virtually any position along the DNA template. However, the main drawback of the technique lies in the fact that each transcription step inherently leads to material loss, and therefore only a limited number of steps can be performed without losing too much signal. Pause sites are often located beyond the reach of stepwise transcription reactions (>75 nt), as observed for the *thiC* riboswitch. Because of this technical limitation, we have sought to develop the NPOM approach allowing to "jump" or "fast-forward" the RNAP at sites located far downstream the sequence, thus allowing to specifically halt the RNAP at mostly any desired positions. To acknowledge this important point, we now have added this sentence in the Discussion (line 449): "Furthermore, although stepwise transcription reaction could in principle be used to specifically position RNAP at any site, the loss of material at each step precludes its use to study pause sites located remotely from the transcription start site (~75 nt)."

Regarding the use of stepwise transcription as a positive control to halt ECs, the Referee is right in mentioning that this technique has been used in several instances and could have been used as a positive control. However, in this paper, because we are developing a novel roadblock, we wanted to compare it with another well-known transcriptional roadblock, i.e., the biotin-streptavidin complex.

Ref2, Major Concern 5.

Referee 2: Figure 3 – removal of NPOM is relatively slow in comparison to the rate of transcription kinetics and appears to be variable between single RNA molecules (e.g., 60% RT vs 40% photo-blocked

at 4 minutes), in bulk – can the authors comment on why this is the case, and what limitations this might bring to the experimental interpretation?

Response by Authors: Removal of NPOM is indeed variable among ECs, as its removal is probabilistic. At each time interval of UVA irradiation, all NPOM moieties have a chance to absorb a photon, therefore resulting in a variable amount of time before all NPOM groups are removed. Since removal is not instantaneous and appears to be relatively slow, it is crucial when monitoring pause half-life to wash unincorporated NTPs before UVA irradiation, thus ensuring that restarting RNAP are completely synchronized. In cases where NPOM removal is not optimal for a given EC, it would result in a lower fraction of ECs resuming transcription elongation. Importantly, this variation should not introduce any unwanted bias in the analysis. Therefore, in the submitted version of the manuscript, it is clearly mentioned that NTPs are washed when monitoring pause half-life, and we do not think that it is important to specifically address the NPOM removal variability.

Ref2, Major Concern 6.

Referee 2: In the Supplemental Methods section on “simulation of transcriptional pause half-life”, it is unclear why the authors are attempting to fit their simulations with a single-exponential decay, when they clearly can describe their data with an analytical solution describing a two-step kinetic model. It seems fairly obvious that the values derived would not be the same in this case, and since the authors are knowingly fitting the data with a function that they know does not describe them well, they should consider switching to a more appropriate double-exponential.

Response by Authors: We thank the Referee for allowing us to clarify this important point. In our manuscript, we are trying to show how NPOM can be used to simplify the study of transcriptional pausing in a complex system containing multiple consecutive pause sites. We agree that in a transcriptional system exhibiting two pause sites, such as described by the theoretical example in Figure 1C, it would be appropriate to employ a two-step kinetic model to analyze the data. However, the goal of this example is to clearly show that single-exponential decay may deceive the user when analyzing such systems containing multiple pause sites. Indeed, by specifically altering the incoming rates of ECs, the example shows that ECs escaping the downstream pause site exhibit different apparent half-lives. Importantly, we agree with the Referee that more information about fitting alternatives should be mentioned in the manuscript to avoid that readers are under the impression that such systems cannot be analyzed using more sophisticated tools. To do so, we have clarified in the manuscript that the example shown in Figure 1C is based on the use of a single-exponential decay (line 58): "It is anticipated that variations (10- to 30-folds) in the rate of incoming EC could significantly affect the apparent half-life of downstream pause sites when fitting the data with a single-exponential decay (Fig. 1c; Supplementary Methods)." Next, we have added a paragraph in the Supplementary Methods in the section Simulation of transcriptional pause half-life. We have added the following (line 111): "The analysis of transcriptional pausing may be achieved by fitting the fraction of escaping ECs using single or multiple exponential decays¹⁻³. It is expected that ECs transiting via a unique transcriptional pause are characterized using a single kinetic constant, which is dictated by the strength of the pause. However, in the case where upstream pause sites are also present, it is likely that multiple kinetic constants are involved due to asynchronous transcription elongation, which could affect the apparent half-life obtained using single-exponential decay. To demonstrate this aspect, we provide here a theoretical example showing that variations in the rate of incoming EC may affect the apparent rate of downstream pause sites obtained when fitting the data with a single-exponential decay."

Ref2, Minor Concern 1.

Referee 2: Can the authors comment on why they see 73% of restart activity in 3C, but only 48% in 3D?

Response by Authors: We agree that the restart activity observed Fig. 3D is significantly lower than what observed in Fig. 3C. While we typically monitored restarting activity ~75% or higher, we decided to use the gel shown in Fig. 3D for the manuscript since it was clearly showing that NPOM-blocked ECs are located at position G89. We believe that inherent variability occurring during the G50 washing steps may have inactivated stalled ECs, therefore resulting in the data shown in Fig. 3D. We regularly observe that purifying processes using G50 columns may inhibit to various degree EC restarting compared to other processes not requiring washing steps. To clarify this point, we have added this (line 208): "Although more work is required to characterize this process, G50 columns used for washing steps during EC preparation may lead to a fraction of complexes not resuming transcription elongation (data not shown)".

Ref2, Minor Concern 2.

Referee 2: Can the authors estimate the extent of pyrimidine dimer formation after 8 minutes of UVA exposure?

Response by Authors: Based on our assays, we estimate that ~85% of ECs resume elongation after 8 min of UVA irradiation (line 184), indicating that less than 15% may be inactivated by UVA light. As implied by the Referee, cyclobutane thymine dimers have been shown to strongly inhibit transcription elongation by the *E. coli* RNAP (Pupov et al., BBRC 2019). However, as shown in Supplementary Fig. 3, there is no intermediate bands appearing when increasing the irradiation time, suggesting that dimer formation does not prevent to a large extent transcription elongation. It is therefore difficult to determine the extent of pyrimidine dimer formation in our current experimental assays.

Ref2, Minor Concern 3.

Referee 2: It would be useful to discuss why T7 RNAP could not be restarted, but *E. coli* transcription could.

Response by Authors: As addressed by another Referee, the lack of T7 RNAP restart possibly suggests that the enzyme does not remain stably bound when encountering NPOM. To discuss this point, we have added this part at the end of the Discussion (line 457): "Our study also indicates that although NPOM can be used to prevent elongation for a variety of polymerases, NPOM removal does not always allow to resume elongation as observed for the T7 RNAP. Further studies will be required to specifically address the interaction between T7 RNAP and NPOM, which could either result in ECs being unstable or blocked in an inactive conformation."

Ref2, Minor Concern 4.

Referee 2: In lines 131-133, the authors state that transcriptional readthrough is not observed, however, in Figure 2B there appear to be faint bands at the full-length size.

Response by Authors: As indicated by the Referee, faint bands are observed where the full-length species are expected (lanes 2 to 4). However, the quantification indicates that the proportion of those bands varies from ~1 to 3 % across the experiments, which is why we previously mentioned that they are not apparent. To remove any confusion, we have changed the text to the following (line 135): "Furthermore, only ~1% to ~3% of transcription readthrough could be observed when incubating transcription reactions for longer periods up to 1 h (Fig. 2b)."

Ref2, Minor Concern 5.

Referee 2: In line 172 it is at first not clear that transcription was performed subsequent (not in parallel) to UVA exposure – this should be clarified.

Response by Authors: The sentence was changed to (line 180): "We first performed a time-dependent exposure of NPOM-containing DNA templates to UVA light and subsequently used the resulting..."

Ref2, Minor Concern 6.

Referee 2: On line 235, the authors say they monitor the “kinetics of transcription elongation”, however, there is no information here describing transcription elongation – only transcriptional pausing. This language should be clarified.

Response by Authors: We have added a brief description about how these experiments were performed. In the Results section, we have added these sentences (line 249): "In these experiments, we first generated EC20 complexes by omitting UTP in the transcription reactions. Following a washing step, we then added NTPs and monitored transcription elongation at different time points."

Ref2, Minor Concern 7.

Referee 2: On line 414 it is debatable whether the claim that NPOM can be “selectively” removed since only a single NPOM can be removed so that removal of this NPOM is not selective over any others. “Selectivity” implies a comparative selection of one site over another, which is not the case here.

Response by Authors: We agree with the Referee and the word "selectively" has been removed. The sentence is now (line 443): "Compared to other EC synchronization methods, NPOM is a unique transcriptional roadblock as it can be removed ..."

Ref2, Minor Concern 8.

Referee 2: Figure 4 – in the text it is difficult to pick apart what the purpose of the “full length” vs. “minimal length” constructs are. The authors mention Figure 4a in line 229, but do not describe the minimal length transcript until line 265. If the minimal length construct were described earlier in the text, it would increase the readability of this section.

Response by Authors: To make the text clearer, we have added a description of the minimal construct earlier in the Results section (line 247): "We also constructed a minimal template not containing most of the sequence located upstream of the *his* RNA hairpin (Fig. 4a, minimal length)."

Ref2, Minor Concern 9.

Referee 2: Many of the Figure cartoons seem a bit primitive (in the era of Biorender and Photoshop); in particular, Figures 1A, 3A, 5A would benefit from a facelift to a more sophisticated representation of the DNA, RNAP and RNA.

Response by Authors: As recommended by the Referee, we have now redrawn the figures to give them a look that feels generally more modern.

Ref2, Minor Concern 10.

Referee 2: On line 52, the authors say “the half-life of pause site is directly affected”. This should read as “the half-life of a pause site...”.

Response by Authors: This has been corrected.

Ref2, Minor Concern 11.

Referee 2: On line 166 it should read: “taking place within”.

Response by Authors: This has been corrected.

Ref2, Minor Concern 12.

Referee 2: On lines 174/175 should read: “UVA illumination...leads”.

Response by Authors: This has been corrected.

Ref2, Minor Concern 13.

Referee 2: On line 183 it should read: “In addition to efficiently blocking...”.

Response by Authors: This has been corrected.

Ref2, Minor Concern 14.

Referee 2: On line 279 it may be better to specify “specifically slows pause escape by”.

Response by Authors: This has been corrected.

Ref2, Minor Concern 15.

Referee 2: On line 280. Delete the first use of “previous”.

Response by Authors: This has been corrected.

Ref2, Minor Concern 16.

Referee 2: On line 420, a hyphen can be added to the word “cryo-electron”.

Response by Authors: This has been corrected.

Ref2, Minor Concern 17.

Referee 2: NPOM’s full UIPAC name should be given upon first introduction both in the Abstract and main text.

Response by Authors: This has been corrected (lines 26 and 81).

Ref2, Minor Concern 18.

Referee 2: It should always be “fitting a function to data” or “fitting data with a function” rather than the other way around.

Response by Authors: This has been corrected.

****Referee 3****

Nadon et al. report use of a photolabile thymidine analog (6-nitropiperonyloxymethyl-thymidine) to roadblock RNA polymerase and then release it for kinetic analyses of transcriptional pausing by photogeneration of the thymidine base. This method of halting and releasing RNA polymerase offers distinct advantages over other available methods to halt RNA polymerase and its use for this purpose has not been reported previously. The authors validate the method by performing kinetic analyses on the well-characterized E. coli his leader pause site, replicating several well-known features of this site but also finding evidence that the nature of the initial halt site upstream from a pause or transcription of the region between an initial halt site and the pause site can affect the apparent dwell time of RNAP at the pause site. The authors also use the method to test the previous finding that altering a pause site upstream from a pause in the thiC leader region can affect the apparent dwell time of RNAP at the downstream pause site, but that this effect disappears when RNAP is restarted at the second pause site using the NPOM roadblock technique. The results will be of broad interest to researchers in the field of transcription, both prokaryotic and eukaryotic, and merit publication. Although new insights into transcriptional regulatory mechanisms from the present study are limited, the significance of work in establishing the utility of a

robust new method is high. Nonetheless, there are a few aspects of the work that could be improved either by increasing the clarity of the manuscript or adding information. These suggestions are listed below.

Ref3, Major Concern 1.

Referee 3: The NPOM method is the most exciting and impactful aspect of this work, but it is surprisingly under described. I could not find a description of the temperature at which the photorestitution reactions were conducted or an estimate of the photon flux used. These basic aspects of the method should be highlighted and systematically described. Even if the authors are unable to measure photon flux, a detailed description of the UV source used should be included (what model of UV lamp was used and what fraction of the 8W power is actually directed to the sample?). Did the authors explore use of different temperatures or UV doses to investigate whether an optimum exists for high levels of T generation with lower levels of transcription complex inactivation? What is the mechanism of EC inactivation during the photolysis reaction? Is it due to dissociation of ECs during the photolysis reaction or some other mechanism? Does addition of GreA/B release a greater fraction of ECs after the photolysis treatment, indicating arrest backtracking as the mechanism? At a minimum, a complete description of reaction conditions and a description of whether complex inactivation increases with time of UV exposure should be included.

Response by Authors: We thank the Referee for allowing us to provide more details about the NPOM cleavage reaction. We have added additional information about the temperature, model of UV lamp and UV tube in the section "NPOM cleavage by UVA irradiation". We were not able to reliably approximate the photon flux in our assays and we therefore did not address this aspect. Also, although we did not test different temperatures, we have used different times of UVA exposure as described at the beginning of our manuscript. Lastly, we agree with the Referee that the mechanism of EC inactivation during photolysis is important and merits to be characterized. However, we did not address this aspect in the current study which was primary aimed at establishing a removable transcriptional roadblock. We therefore do not have data regarding EC inactivation or the effect that GreA/B could have on inactivated complexes.

Ref3, Major Concern 2.

Referee 3: The authors report that reducing the length of DNA upstream of the *his* pause causes a reduction in the apparent half-life, but they do not report the rate at which RNAP transcribes from the initial halt site to the pause site for the two templates studied. At a minimum, the authors should show the portions of the gels with the halted complexes in the figures. It seems possible the difference could result from differences in the rate of escape from the initially halted complexes rather than the length of DNA transcribed. This doesn't directly bear on the utility of the NPOM method, but is important to ensure the authors aren't mis-describing their results.

Response by Authors: As indicated by the Referee, it is possible that the half-lives of the *his* pause in the full- and minimal lengths are affected by different rates of escape from ECs leaving their respective initial halt position. After examining the original gels, we came to the conclusion that it would be hard to have a definite answer about this question. As an example, here are two of the best gels that we got for the full-length and minimal length constructs in the absence of NPOM. (*add some explanation here*). However, based on the half-lives obtained using the minimal length construct with and without NPOM, it seems that the initially halted ECs (- NPOM) do not exhibit a significant difference in rate escape compared to ECs leaving the pause site (+ NPOM), suggesting that this might not be a widespread phenomenon. Nevertheless, to avoid any misinterpretation about this, we have added this sentence in the text (line 285): "Since the half-lives calculated for the full-length and minimal constructs are determined using different initially halted complexes (EC20 vs EC102), it is possible that the respective rate of escape of these complexes may contribute to some extent to the observed differences in pause half-life".

Ref3, Major Concern 3.

Referee 3: The failure to see a NusA effect on the *his* pause using the NPOM method is confusing and the explanation offered does not accord with prior demonstrations that NusA enhances pausing at the *his* pause by ECs halted just 1 nt before the pause site (e.g., Guo et al., 2018 Mol. Cell 69:816). Perhaps some sort of mixing experiment could be performed to test if the NusA is active on other complexes in the same solution. Given prior findings, it seems possible that some sort of experimental artifact has arisen in this experiment.

Response by Authors: We are not entirely sure what the Referee means by "a failure to see a NusA effect". In the Supplementary Table 1, the half-lives of the *his* pause (full-length construct with NPOM, *pLacUV5-his-132-NPOM (FL)*) in the absence and presence of NusA are 12 ± 2 s and 44 ± 5 s, respectively. Furthermore, in the context of the minimal construct with NPOM (*pLacUV5-his-62-132 NPOM (ML)*), we observed that half-lives of 12 ± 2 s and 18 ± 2 s without and with NusA. Although the NusA effect appears smaller in the minimal construct, the results obtained with both constructs suggest that NusA is functional in our assays and provides an increased half-life to the *his* pause.

Ref3, Major Concern 4.

Referee 3: Although the authors mention how halting ECs with NPOM allows tests of the effects of upstream events on pausing, the descriptions focus primarily on pause kinetics and don't highlight other possible applications such as preparing ECs for structural studies in arbitrary sequence contexts using RNA synthesis rather than direct reconstitution. I think highlighting some other applications as well as making the point that the method allows a formal distinction between the properties of ECs actively transcribing through a template position vs. restarting after being halted artificially at a template position could raise the impact of the manuscript. The authors should consider adding a section to the discussion making these points.

Response by Authors: We thank the Referee for allowing us to expand on the additional applications where NPOM could be used. We have added this text in the Discussion (line 445): "Importantly, in contrast to other approaches relying on conventional DNA templates, the use of NPOM-containing templates allows to monitor specific properties of ECs restarting from a given position along the DNA template. For example, NPOM could be used in a vast array of biochemical assays requiring the use of stable ECs, such as single-molecule fluorescence, RNA-protein cross-linking and cryo-electron microscopy studies".

Ref3, Minor Concern 1.

Referee 3: The description of current methods for measuring pause dwell times as being biased seems misleading. An extensive literature already recognizes the difference between apparent pause dwell times and the intrinsic kinetics of pausing. Although the authors at times use more appropriate descriptions of "apparent half-lives" rather than suggesting that unsynchronized methods are being used to measure actual dwell times, they should be careful to always make this distinction (e.g., line 57 should refer to apparent half life not half-life) and not to suggest that the distinction between apparent dwell times and actual rates of pause escape is not already well established in the transcription literature. Indeed in a paper the authors should cite (e.g. at line 53), Theissen et al., 1990 (Anal. Biochem. 189:254) thoroughly describe this distinction as well as method to quantify pause strengths independent of asynchrony effects. In addition, true pause escape rates can be elucidated by kinetic modeling, and that approach also has been used extensively. My recommendation would be to avoid the use of "bias" as a descriptor and simply make that point that asynchronous transcription is well known to cause apparent dwell times to be longer than actual pause escape rates – a problem the authors method can deconvolute.

Response by Authors: We thank the Referee for the suggested reference, which has been added at the line 53. We have also rewritten some parts of the manuscript to avoid the use of the word "bias". An

example of this is found in the abstract (line 24): "However, this approach may produce apparent half-lives that are longer than true pause escape rates in biological contexts where multiple consecutive pause sites are present".

Ref3, Minor Concern 2.

Referee 3: The penultimate sentence of the abstract is confusing. It's unclear if the authors mean the methods are contrasting or the results are contrasting since the sentence suggests a method is in contrast with a result. Consider revising this sentence.

Response by Authors: We have rewritten this sentence as follows: "We find that an RNA hairpin structure located upstream to the region affects the half-life of the 5' most proximal pause site—but not of the 3' pause site—in contrast to results obtained using conventional approaches not preventing asynchronous transcription".

Ref3, Minor Concern 3.

Referee 3: In the introduction, the authors should consider adhering to scientific convention by describing already established (published) findings in the present tense rather than in the past tense.

Response by Authors: This has been corrected.

Ref3, Minor Concern 4.

Referee 3: li60 It is misleading to suggest that conventional methods lead to biased conclusions, especially without citing an example, as it implies this distinction between true pause half-life and apparent pause half-life has not previously been appreciated, which is not the case. It would be more accurate to say unsynchronized transcription through a pause site can cause apparent dwell times to be greater than an accurate measurement of pause escape rate would reveal.

Response by Authors: We have rephrased this sentence as follows (line 61): "Importantly, multiple consecutive pause sites are observed in several biological systems, suggesting that asynchronous transcription in such contexts may complicate the determination of the actual rate of pause escape and therefore the biological relevance of transcriptional pause sites".

Ref3, Minor Concern 5.

Referee 3: li72 and other locations (e.g. li377, 385, 386) – "...enzyme...allows to block...elongation." There are multiple uses of this incorrect grammatical construction, which combines an active verb with an infinitive, in the manuscript. This example could be reworded more simply as "...enzyme...can block...elongation." or "...enzyme...can be used to block...elongation." The other cases should also be reworded to avoid confusing readers about the authors' intended meaning.

Response by Authors: This has been corrected.

Ref3, Minor Concern 6.

Referee 3: li89 – delete "Clearly," At the start of the sentence.

Response by Authors: This has been corrected.

Ref3, Minor Concern 7.

Referee 3: li140 – The authors suggest that NPOM prevents transcription elongation by inhibiting binding of ATP to NPOM-T in the template DNA strand. It is also possible that NPOM-T fails to translocate into the active site. Failure to translocate would cause RNAP to stall 1-bp earlier on the DNA template than if NPOM-T prevents ATP binding after translocation but would give the same RNA 3' end in the halted

transcription complex. It would be ideal if the authors could distinguish and report on these two possibilities, for example by using exoIII footprinting or by testing for the sensitivity of the halted complex to pyrophosphate. At a minimum, the authors should mention both possibilities.

Response by Authors: As suggested by the Referee, we have addressed the two possibilities in the text (line 145): "Alternatively, it is also possible that NPOM-T fails to translocate into the active site, which would result in transcripts having the same 3' end in halted complexes".

Ref3, Minor Concern 8.

Referee 3: li166 – The authors show that NPOM-halted transcription complexes are less prone to backtracking than streptavidin-halted transcription complexes. It is unclear, however, if NPOM-T inhibits backtracking or if streptavidin promotes backtracking. The authors could test whether NPOM-T actively inhibits backtracking by halting RNAP at a site known to be susceptible to backtracking and using Gre cleavage or exoIII footprinting to assess the susceptibility to backtracking of complex halted by NPOM-T vs. those halted by NTP-deprivation (eg, because the reaction mix lacks ATP). Although this experiment is not required for publication of the authors work, it would enhance their paper. At a minimum, the authors should mention that their result could be explained either because NPOM-T inhibits backtracking or because streptavidin promotes backtracking.

Response by Authors: As indicated by the Referee, our assays do not address the fact that streptavidin may promote EC backtracking since we do not have a control where backtracking was assessed in the absence of streptavidin. We have rewritten this sentence to mention the fact that streptavidin could promote backtracking (line 173): "Importantly, although these results show that NPOM and streptavidin produce different backtracking effects, further experiments are required to determine whether backtracking is either inhibited with NPOM or promoted with streptavidin".

Ref3, Minor Concern 9.

Referee 3: li200 – The authors report that a fraction of ECs are not in an active conformation after photolysis of NPOM. As described in major point 1 above, the authors should attempt to determine an explanation for this inactivation. At a minimum, the authors should offer possible explanations for the inactivation.

Response by Authors: Multiple causes could explain the varying degrees of transcription restart after UVA irradiation and NPOM removal. For example, in the course of our studies, we have observed when performing stepwise transcription that G50 washing steps could prevent ECs from restarting. Because G50 washing steps are used in the present study, we think that they could be causing EC inactivation. To acknowledge this point, we have added this part in the Results (line 208): "Although more work is required to characterize this process, G50 columns used for washing steps during EC preparation may lead to a fraction of complexes not resuming transcription elongation (data not shown)".

Ref3, Minor Concern 10.

Referee 3: li211 – The authors report that T7 RNAP halted by NPOM does not resume transcription after NPOM photolysis. T7 RNAP is known to form relatively unstable ECs. Is the failure to resume transcription due to T7 RNAP dissociating from DNA during the photolysis reaction or to some other form of inactivation? If T7 RNAP is halted at the same site by simple NTP deprivation, is it equally unable to restart, or does the photolysis reaction inactivate the halted T7 RNAP?

Response by Authors: We did not investigate the effects of NTP deprivation or photolysis reaction on T7 RNAP. To address this point, we have added in the Results section (line 221): "Additional work will be needed to determine whether stable T7 RNAP complexes are obtained through NPOM roadblocking or if such complexes are unstable".

Ref3, Minor Concern 11.

Referee 3: li306 – should be “apparent half-life”.

Response by Authors: This has been corrected.

Ref3, Minor Concern 12.

Referee 3: li374 – “...NPOM may also be used as a roadblock for T7 RNAP...”. This statement is misleading because the authors have not established that T7 RNAP remains on DNA after encountering the NPOM roadblock.

Response by Authors: We thank the Referee for mentioning this point. It is true that we do not provide data showing that T7 RNAP remains stably bound to the DNA when roadblocked by NPOM. The sentence has been rewritten as such (line 403): “Furthermore, we show here that NPOM may also be used as a roadblock for Taq and Phusion DNAPs”.

Ref3, Minor Concern 13.

Referee 3: li391 delete “s” from “investigations”.

Response by Authors: This has been corrected.

Ref3, Minor Concern 14.

Referee 3: Methods section – The authors do not describe the sources of NPOM oligos, RNAP, NusA, Gre factors, or other enzymes used in the studies. This information must be included to allow enable reproduction of the authors work. For a paper in which an important new method is being reported, I found the Methods section surprisingly sparse on information. The methods for NPOM cleavage (li452) should provide all relevant information.

Response by Authors: All relevant information about the purification of *E. coli* RNAP, σ^{70} and NusA was added in the Methods section. The information about T7 RNAP and GreB was added in the Supplementary Methods section. We also have also improved the methods describing the NPOM cleavage.

REVIEWERS' COMMENTS:

Reviewer #2 (Remarks to the Author):

In the manuscript by Nadon et al., the authors present a significant advance in the field, wherein they engineered a photolabile modification, NPOM, to study pausing of transcription elongation complexes. The authors demonstrate clearly that NPOM creates a stable "roadblock" for bacterial RNA polymerase (RNAP), which then is releasable upon excitation at 365 nm. They further show that this roadblock does not cause backtracking, as demonstrated by lack of cleavage product in the presence of transcription factor GreB. Lastly, the authors use this technology to study the thiC riboswitch, which contains multiple pause sites, the latter of which cannot be described by single-exponential decay functions. Overall, the manuscript presents a clear narrative that is of scholarly character and will be of great interest to the transcription field.

The authors have done a thorough job addressing all three reviewers' concerns, including performing a more thorough kinetic analysis with the help of two new collaborators/co-authors. All my concerns have been resolved satisfactorily.

Reviewer #3 (Remarks to the Author):

The authors have done a good job responding to the reviewers' critiques and revising their manuscript to fix the issues noted by the reviewers. I find that the revised manuscript is suitable for publication. Of course, the authors should be given a chance to make final edits before the manuscript is processed to for publication.